# A Multifaceted benchmarking of synthetic electronic health record generation models

Chao Yan [1,7], Yao Yan [2,7], Zhiyu Wan [1,7], Ziqi Zhang [3], Larsson Omberg [2], Justin Guinney [4,5], Sean D. Mooney [4,8] ✉ & Bradley A. Malin [1,3,6,8] ✉

Synthetic health data have the potential to mitigate privacy concerns in supporting biomedical research and healthcare applications. Modern approaches for data generation continue to evolve and demonstrate remarkable potential. Yet there is a lack of a systematic assessment framework to benchmark methods as they emerge and determine which methods are most appropriate for which use cases. In this work, we introduce a systematic benchmarking framework to appraise key characteristics with respect to utility and privacy metrics. We apply the framework to evaluate synthetic data generation methods for electronic health records data from two large academic medical centers with respect to several use cases. The results illustrate that there is a utility-privacy tradeoff for sharing synthetic health data and further indicate that no method is unequivocally the best on all criteria in each use case, which makes it evident why synthetic data generation methods need to be assessed in context.

The analysis of large quantities of data derived from electronic health records (EHRs) has supported a number of important investigations into the etiology of disease, personalization of medicine, and assessment of the efficiencies and safety in healthcare administration[1–3]. Mounting evidence suggests that broader data sharing would ensure reproducibility, as well as larger and more robust statistical analysis[4,5]. Yet EHR data are rarely shared beyond the borders of the healthcare organization that initially collected the data. This is due to a number of reasons, some of which are technical, others of which are more social in their nature. In particular, the privacy of the patients to whom the data correspond is often voiced as a reason for not sharing such data[6,7].

Over the past several years, the notion of synthetic versions of EHR data has been proposed as a solution for broader data sharing[8,9]. While synthetic data are not novel in principle, recent advances in machine learning have opened up new opportunities to model complex statistical phenomena within such data, which could support a variety of applications[8]. For instance, detailed synthetic data could make it easier to develop techniques for clinical decision support, as

well as prototype automated research workflows[10]. At the same time, synthetic data can sever the direct relationship with the real patient records upon which they are based, thus mitigating privacy concerns[9,11]. As a result, a growing set of research initiatives have developed, or are considering the use of, synthetic data sharing, including the National COVID Cohort Collaborative (N3C)[12] (https://covid.cd2h.org/n3c) supported by the U.S. National Institutes of Health and the Clinical Practice Research Datalink[13] (https://cprd.com/synthetic-data) sponsored by the U.K. National Institute for Health and Care Research.

While various synthetic EHR data generation techniques have been proposed, generative adversarial networks (GANs) have gained a substantial amount of attention, with their potential illustrated for a wide range of applications[8,14,15]. Informally, a GAN is composed of two neural networks that evolve over a series of training iterations: (1) a generator that attempts to create realistic data and (2) a discriminator that aims to distinguish between synthetic and real data[16]. Iteratively, the generator receives feedback from the discriminator, which it

[1]Department of Biomedical Informatics, Vanderbilt University Medical Center, Nashville, TN, USA. [2]Sage Bionetworks, Seattle, WA, USA. [3]Department of Computer Science, Vanderbilt University, Nashville, TN, USA. [4]Department of Biomedical Informatics and Medical Education, University of Washington, Seattle, WA, USA. [5]Tempus Labs, Chicago, IL, USA. [6]Department of Biostatistics, Vanderbilt University Medical Center, Nashville, TN, USA. [7]These authors contributed equally: Chao Yan, Yao Yan, Zhiyu Wan. [8]These authors jointly supervised this work: Sean D. Mooney, Bradley A. Malin. ✉e-mail: sdmooney@uw.edu; b.malin@vumc.org

leverages to tune its network to more effectively imitate the real data. Unlike traditional data synthesis approaches, which either explicitly model clinical knowledge or make assumptions about the relationships between features[17,18], models built through GANs circumvent these challenging issues by directly learning complex relationships from multi-dimensional data.

The field has advanced rapidly; however, there has been little attention paid to benchmarking, which is a concern for several reasons. First, there is a lack of consensus on the evaluation metrics that should be applied to assess synthetic EHR data. Yet this is critical to comprehensively compare and contrast candidate synthesis models. While several investigations have demonstrated the superiority of new GAN models over existing models, the comparisons are not systematic and are susceptible to the self-assessment trap[19] in that the model developers benchmark their own models. Second, there is a wide range of use cases for synthetic data, each with its own set of priorities regarding what aspects of the data should be preserved. Most publications on EHR data synthesis neglect the use case, such that it is unclear what conditions are ideal for the simulation model. Third, prior evaluations have typically focused on the simulation and evaluation of a single execution, such that only one synthetic dataset is generated by each model[20,21]. This is problematic because GAN models are often associated with unstable training trajectories, which can lead to quite different models and inconsistencies in the quality of the generated data[15,22,23].

In this paper, we introduce a benchmarking framework to evaluate GAN models for EHR synthesis (Fig. 1). We focus specifically on structured EHR data[24], as this type of data has supported numerous clinical association and outcome prediction studies. There are several specific contributions of this work:

1. We incorporate a complementary set of data utility and privacy metrics into the benchmarking framework to enable a systematic evaluation of synthesis models.

2. We introduce a rank-based scoring mechanism that converts scores from individual metrics into the final score of a model. This mechanism enables tradeoffs between competing evaluation metrics.

3. To enable broad reuse, the framework accommodates multiple aspects of complexity from GAN-based synthesis, including various data types in real data (e.g., categorical and continuous feature representations), different data synthesis paradigms (i.e., the manner by which real data is applied to train generative models), and the inconsistent quality of synthesized data by GAN models.

4. We use EHR data from two large academic medical centers in the United States to benchmark the state-of-the-art GAN models. We demonstrate the flexibility and generalizability of the framework through contextualized construction of concrete use cases, where synthetic EHRs already (or have the potential to) provide support.

5. Our findings show that no model is unequivocally the best on all criteria in each use case for each dataset. This result clearly illustrates why synthetic data generation models need to be systematically assessed in the context of their use case before their application.

## Results

### Benchmarking framework

The benchmarking framework focuses on two perspectives—utility and privacy (Fig. 1). Table 1 summarizes the metrics incorporated into the Multifaceted assessment phase of the benchmarking framework. The "Methods" section provides the details for these metrics.

In this study, data utility was measured through the principles of resemblance and outcome prediction. Resemblance focuses on the internal distributional characteristics of data and considers two complementary criteria: (1) feature-level statistics—the ability to capture characteristics of the distributions of real data—through dimension-wise distribution[25], column-wise correlation[26], and latent cluster analysis[27], and (2) record-level consistency, in terms of both the ability to generate individual records that comply with clinical knowledge as measured by clinical knowledge violation and the ability to capture the quantity of record-level information by medical concept abundance. By contrast, outcome prediction focuses on downstream modeling tasks and measures the ability to train and evaluate machine learning models as measured by model performance[28] and feature selection. Notably, the three metrics for feature-level statistics differ in the distributions considered and correspond to the marginal distribution, the correlation between two features, and the joint distribution of all features, respectively.

Data privacy was assessed through four metrics: (1) attribute inference risk[25], in which unknown attribute values of interest are predicted from a set of known attribute values, (2) membership inference risk[25], which indicates whether a real record was used to train a generative model, (3) meaningful identity disclosure risk[29], in which the identity and sensitive attributes of a patient's record are detected as being part of the real dataset, and (4) nearest neighbor adversarial accuracy (NNAA) risk[30], which indicates whether a generative model overfits the real training data. The privacy risks were measured under the typical assumption about an attacker's knowledge. Specifically, it was assumed that the attacker has access to synthetic data, but not the generative models[20,25,31]. It should be noted that we will mainly use the term attribute (instead of feature) in the context of privacy risk to be consistent with the privacy literature. In addition, we will use record to denote the data for a patient.

Prior studies into synthetic EHR data compared models using only a single training of the generator. Yet this can lead to a less reliable evaluation because GAN models can be unstable relative to training (leading to models with large differences in parameters)[15,22], making it hard to yield synthetic datasets of consistent data quality[32–34]. In this work, we integrated a mechanism into the framework that involves training multiple models and generating data from each to capture variations in models and baking biases into model comparison (Synthetic EHR data generation phase in Fig. 1). Specifically, we performed model training and synthetic data generation five times for each model. We then selected the three datasets that best preserved the dimension-wise distribution of real data for the benchmarking analysis. We relied upon this metric to filter synthetic datasets because it is a basic utility measure and provides face-value evidence of the usability of synthetic data. In doing so, we dropped the synthetic datasets that poorly captured the first-moment statistics of individual features.

We designed a ranking mechanism that scores each model based on the results of three independently generated datasets. Specifically, for each metric, we calculated the metric scores for all of the datasets generated by all candidate models. Next, for each metric, we ranked the synthetic datasets—with smaller ranks denoting better performance of a dataset on the given metric. For each metric, we defined the average of the ranks of each model's three synthetic datasets as the rank-derived score on this metric. Thus, there were twelve lists of ranks and rank-derived scores, one for each metric. The final score for a model was the weighted sum of the rank-derived scores, where the weights were tailored to the specific use case in the Model recommendation phase. An example of this process is provided in "Methods".

We used this framework to evaluate five EHR synthetic data generation models based on GANs[8]: (1) medGAN[25], (2) medBGAN[35], (3) EMR-WGAN[20], (4) WGAN[35], and (5) DPGAN[36]. In addition, we incorporated a baseline approach that randomly samples the values of features based on the marginal distributions of the real data to complement the scope of benchmarking in terms of the variety of model behavior. We refer to this approach as the sampling baseline, or Baseline. Interestingly, as our results illustrate, this approach outperformed the GAN

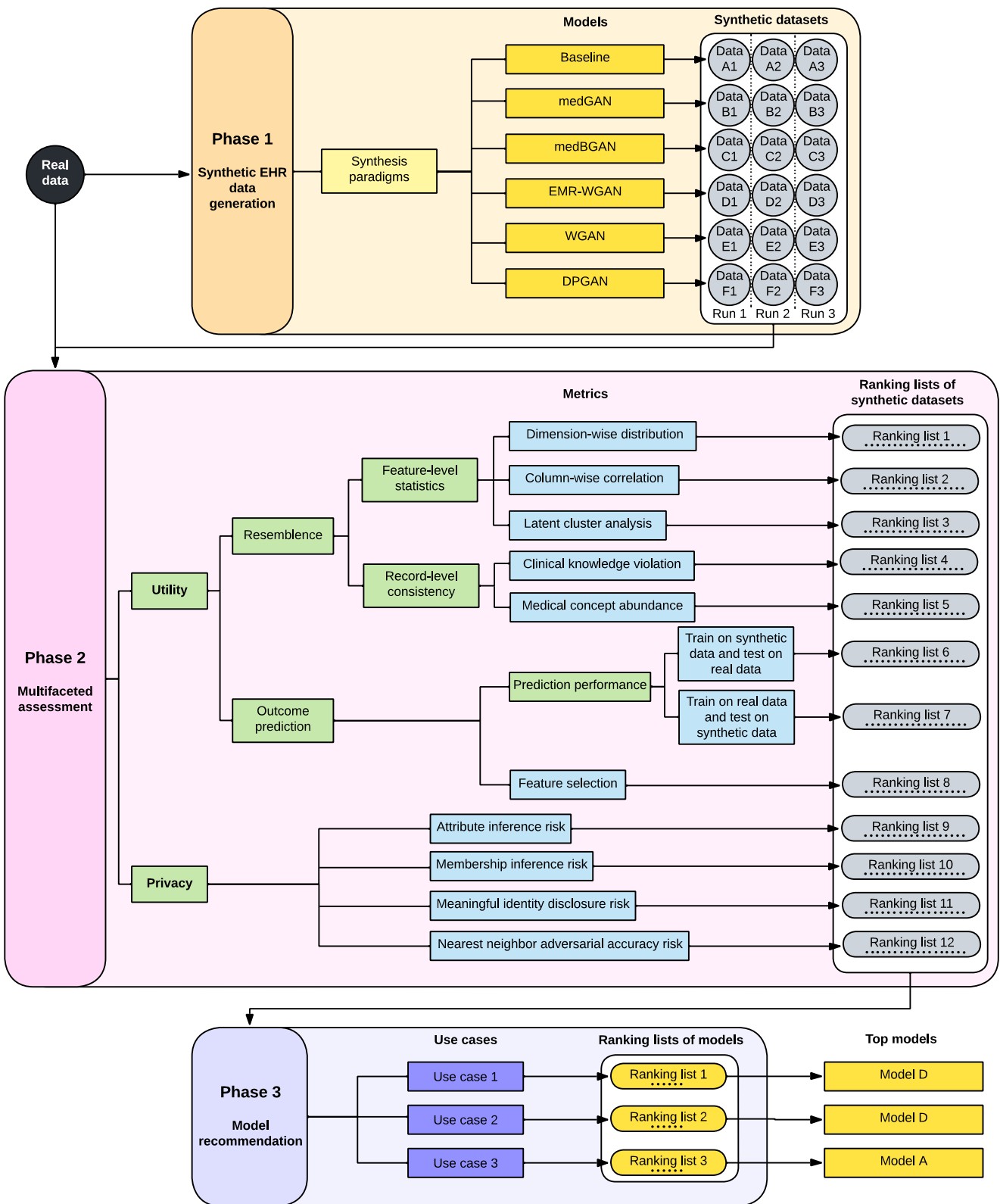

**Fig. 1 | An overview of the synthetic EHR data generation benchmarking framework.** The framework is composed of three phases: (1) a synthetic EHR data generation process, (2) a multifaceted assessment process, and (3) a use case-specific model recommendation process. In Phase 1, given a synthesis paradigm and the real data, we generate multiple (specifically, three in our experiments) synthetic datasets using each data generation model. In Phase 2, each generated synthetic dataset is assessed and assigned a value in terms of each assessment metric. Afterward, all synthetic datasets will be ranked according to their values in terms of each metric. In Phase 3, for each use case, we assign a weight to each metric and convert multiple (specifically, twelve in our experiments) ranking lists of synthetic datasets into one ranking list of models. Finally, the top ranked model for each use case is recommended. EHR: electronic health record.

**Table 1 | A summary of the metrics in the framework**

| | Metric | Summary | Direction |
|---|---|---|---|
| Utility | Dimension-wise distribution | Goal: The ability to capture marginal feature distributions in real data. Measurement: Average of the absolute prevalence differences (APD) for binary features and the average of the Wasserstein distances (AWD) for continuous features between real and synthetic datasets[25]. | ↓ |
| | Column-wise correlation | Goal: The ability to capture the relationship between two features in real data. Measurement: Average of the cell-wise absolute differences of the Pearson correlation coefficient matrices derived from real and synthetic datasets[26]. | ↓ |
| | Latent cluster analysis | Goal: The ability to capture the joint distribution of all features in real data. Measurement: Deviation of a synthetic dataset in the underlying latent space from the corresponding real dataset in terms of unsupervised clustering[27]. | ↓ |
| | Clinical knowledge violation | Goal: The ability to learn the clinical knowledge at the patient level. Measurement: Proportion of generated records that violate clinical knowledge derived from the real dataset (e.g., the synthetic records for male patients are frequently associated with pregnancy diagnosis codes). | ↓ |
| | Medical concept abundance | Goal: The ability to retain record-level information from the real data. Measurement: Normalized Manhattan distance between the distributions of the number of assigned distinct medical concepts for real and synthetic records. | ↓ |
| | TSTR Model performance | Goal: The ability to approximate the performance of the downstream task of machine learning model development. Measurement: Given an outcome prediction task, this is calculated as the model performance, typically the area under the receiver operating characteristics curve (AUROC), in the scenario of training on synthetic dataset and testing on real dataset (TSTR)[28]. | ↑ |
| | TRTS Model performance | Goal: The ability to generate convincing and realistic data records for different labels. Measurement: Given an outcome prediction task, this is calculated as the model performance, typically the AUROC, in the scenario of training on real dataset and testing on synthetic dataset (TRTS)[28]. | ↑ |
| | Feature selection | Goal: The ability to support model interpretability in downstream tasks. Measurement: The proportion of shared important features for models trained on a synthetic dataset and the corresponding real dataset. | ↑ |
| Privacy | Attribute inference risk | Goal: The adversary's ability to infer sensitive attributes of a targeted record. Adversarial knowledge: Demographics and some sensitive attributes of a targeted record. Measurement: The weighted sum of F1 scores of the inferences of other sensitive attributes[20,25]. | ↓ |
| | Membership inference risk | Goal: The adversary's ability to infer the membership of a targeted record. Adversarial knowledge: A set of attributes of a targeted record. Measurement: The F1 score of the inference based on Euclidean distances between the targeted record and all synthetic records[20,25]. | ↓ |
| | Meaningful identity disclosure risk | Goal: The adversary's ability to identify synthetic records with meaningful attributes. Adversarial knowledge: A population dataset with identities. Measurement: The adjusted re-identification risk considering the linkage between the synthetic dataset and the real dataset, the linkage between the synthetic dataset and the population dataset, and the rareness of each sensitive attribute in the real dataset[29]. | ↓ |
| | Nearest neighbor adversarial accuracy risk | Goal: The extent to which a generative model overfits the real training dataset. Measurement: The difference between 1) the aggregated distance between records in the synthetic datasets and records in the evaluation dataset and 2) the aggregated distance between records in the synthetic datasets and records in the real dataset[30]. | ↓ |

The direction of the values indicates if a higher (up arrow) or lower (down arrow) value is better.

**Table 2 | A summary of the GAN models applied for benchmarking**

| Model | Distance measure (loss function) | Auto-encoder for discrete data generation | Normalization | Additional privacy components |
|---|---|---|---|---|
| medGAN[25] | Jensen-Shannon divergence | Yes | BatchNorm for generator | No |
| medBGAN[35] | f-divergence | Yes | BatchNorm for generator | No |
| EMR-WGAN[20] | Wasserstein divergence | No | BatchNorm for generator; LayerNorm for discriminator | No |
| WGAN[35] | Wasserstein divergence | Yes | BatchNorm for generator | No |
| DPGAN[36] | Wasserstein divergence | Yes | BatchNorm for generator | Yes (differentially private stochastic gradient descent) |

All models share the generator-discriminator architecture for EHR data synthesis, but differ in their specializations to enhance either utility or privacy.

models in practical use cases. Table 2 summarizes the key characteristics of these models, while complete details are provided in Methods.

## Datasets
We performed our benchmarking using EHR data from two large academic medical centers in the United States, the University of Washington (UW) and Vanderbilt University Medical Center (VUMC). Table 3 provides summary characteristics for the data. The UW dataset was introduced in a public DREAM Challenge for mortality prediction[37]. It includes 2665 features from 188,743 patients who visited the UW health system between January 2007 and February 2019.

By contrast, the VUMC dataset includes 2592 features from 20,499 patients who tested positive for COVID-19 at an outpatient visit from March 2020 to February 2021.

## Data utility
Figure 2 depicts the prevalence of categorical features for the real and synthetic datasets. It can be seen that Baseline consistently achieved the most similar marginal distributions to real data for both datasets, as it achieves the lowest APD of all synthesis models. Among the GAN synthetic datasets, those generated by EMR-WGAN achieved a clear pattern for both UW and VUMC datasets in which all binary features

**Table 3 | The characteristics of the benchmarking datasets**

| | UW Dataset | | VUMC Dataset | |
|---|---|---|---|---|
| Age | – | | 26.0, 40.3, 55.8 | 41.0 ± 18.7 |
| **Race** | | | | |
| White | 69.9% | 131,830 | 65.2% | 13,366 |
| Black | 7.9% | 14,956 | 8.8% | 1794 |
| Asian | 9.4% | 17,646 | 1.9% | 384 |
| American Indian or Alaska Native | 1.5% | 2836 | 0.0% | 42 |
| Pacific Islander | 0.8% | 1563 | 0.0% | 0 |
| Unknown | 10.5% | 19,912 | 24.0% | 4913 |
| **Gender** | | | | |
| Male | 45.3% | 85,490 | 43.9% | 8990 |
| Female | 54.7% | 103,253 | 56.1% | 11,509 |
| **Medical features for generation** | | | | |
| *Binary features* | | | | |
| # of unique codes | 2662 | | 2581 | |
| Diagnosis (Phecode) | 1736 | | 1269 | |
| Procedure (Category) | 66 | | 67 | |
| Medication (RxNorm Ingredient) | 860 | | 1245 | |
| # of unique codes per patient | 13.0, 30.0, 51.0 | 36.8 ± 31.3 | 6.0, 21.0, 59.0 | 45.3 ± 63.6 |
| *Continuous features* | | | | |
| Diastolic pressure | – | | 68.0, 75.0, 82.0 | 75.0 ± 10.7 |
| Systolic pressure | – | | 114.0, 124.0, 136.0 | 125.3 ± 15.9 |
| Pulse | – | | 77.3, 90.0, 104.3 | 91.4 ± 18.6 |
| Temperature | – | | 36.8, 37.1, 37.7 | 37.3 ± 0.6 |
| Pulse Oximetry | – | | 95.1, 97.1, 99.0 | 97.1 ± 2.1 |
| Respirations | – | | 16.0, 18.0, 23.9 | 19.6 ± 4.4 |
| Body Mass Index | – | | 24.4, 30.3, 38.1 | 31.3 ± 8.7 |
| **Data split for prediction** | | | | |
| *Training data* | | | | |
| Positive label | 3.8% | 4966 | 3.8% | 541 |
| Negative label | 96.2% | 127,158 | 96.2% | 13,808 |
| *Evaluation data* | | | | |
| Positive label | 3.8% | 2129 | 4.2% | 260 |
| Negative label | 96.2% | 54,490 | 95.8% | 5609 |

x,y,z represents the first quartile, median, and third quartile. x ± y represents the mean and one standard deviation. x%y indicates that the percentage of y patients is x% among all patients.

were closely distributed along the diagonal line, suggesting a strong capability of retaining the first-moment statistics in the real data. By contrast, medBGAN, medGAN, and DPGAN (in descending order of APD) were less competitive, exhibiting a tendency to deviate from the marginal distribution in real data. For the VUMC dataset, the synthetic datasets from WGAN exhibited a similar pattern to those generated by EMR-WGAN in terms of APD. Yet for the UW dataset, WGAN achieved a substantially higher APD than EMR-WGAN. DPGAN, which enforced a differential privacy constraint on WGAN, led to heavier deviations

from the marginal distribution in real data. In addition, all models, except for WGAN and DPGAN, achieved higher APD for the UW dataset than for the VUMC dataset.

Figure 3 depicts seven data utility metrics for the real and synthetic datasets. EMR-WGAN exhibited the highest average utility for the UW dataset for all individual utility metrics except for dimension-wise distribution (Fig. 3a–g). EMR-WGAN was also the best model according to five (Fig. 3h, i, k, l, and m) of the seven metrics for the VUMC dataset. For the other two metrics (i.e., clinical knowledge violation and feature selection), WGAN achieved the best performance, which suggests it is more adept at record-level consistency and model interpretability (Fig. 3j, n). By contrast, for the VUMC dataset, Baseline consistently had the worst average utility (Fig. 3h–n). Also, for the UW dataset, Baseline was one of the two worst performing models for six metrics (Fig. 3b–g). DPGAN also performed poorly as it can be seen that it was associated with the lowest utility for the UW dataset in terms of column-wise correlation, latent cluster analysis, medical concept abundance, and feature selection.

## Data privacy

Figure 4 depicts four data privacy metrics for the real and synthetic datasets. All synthetic datasets achieved a lower privacy risk than the real data. In terms of membership inference risk, meaningful identity disclosure risk, and NNAA risk, Baseline posed the lowest average risk except that DPGAN achieved the lowest average meaningful identity disclosure risk for the UW dataset. In terms of attribute inference risk (Fig. 4a, e), WGAN achieved the lowest average risk for the UW dataset and medGAN achieved the lowest average risk for the VUMC dataset. On the other hand, EMR-WGAN posed the highest average risk on all privacy metrics. However, even the highest risks posed by EMR-WGAN in our experiments for each privacy metric (i.e., 0.152 for the attribute inference risk, 0.276 for the membership inference risk, 0.015 for the meaningful identity disclosure risk, and 0.020 for the NNAA risk) can be regarded as low risk if we segment the range of risk from 0 to 1 equally into three categories (i.e., low, median, and high) in which the low risk category means the risk is lower than 0.333. Note that Zhang et al.[20] used the highest risk among risks posed by either medBGAN or WGAN as the threshold for both the attribute inference risk and the membership inference risk (i.e., 0.152 and 0.242, respectively, for our experiments). From this perspective, only EMR-WGAN for the VUMC dataset has a slightly higher membership inference risk than the threshold. In addition, El Emam et al.[29] used 0.09 as the threshold for the meaningful identity disclosure risk to determine whether a synthetic dataset is risky, which is much higher than 0.015, under the guidelines from the European Medicines Agency[38] and Health Canada[39] for the sharing of clinical data, whereas some custodians used 0.333 as the threshold[40,41]. Furthermore, Yale et al.[30] applied 0.030 as the threshold for the NNAA risk, which suggests that 0.020 is at an acceptable level.

## Utility privacy tradeoff

Figure 5a, b provides a summary of the models' rank-derived scores with respect to each utility and privacy metric. The privacy-utility tradeoff in both datasets is evident. A generative model associated with a higher utility (e.g., EMR-WGAN) usually had a lower privacy score, whereas a model associated with a higher privacy score often had lower utility, as illustrated in Baseline for the VUMC dataset and DPGAN for the UW dataset. The other models, which exhibited a moderate utility ranking, were often associated with a moderate privacy ranking as well. This phenomenon generally holds true for all of the models tested.

To illustrate how the benchmarking metrics complement each other, we investigated their pairwise correlation. In doing so, a strong correlation indicates that two metrics provide similar rank-derived scores across models and two real datasets, whereas a weak correlation

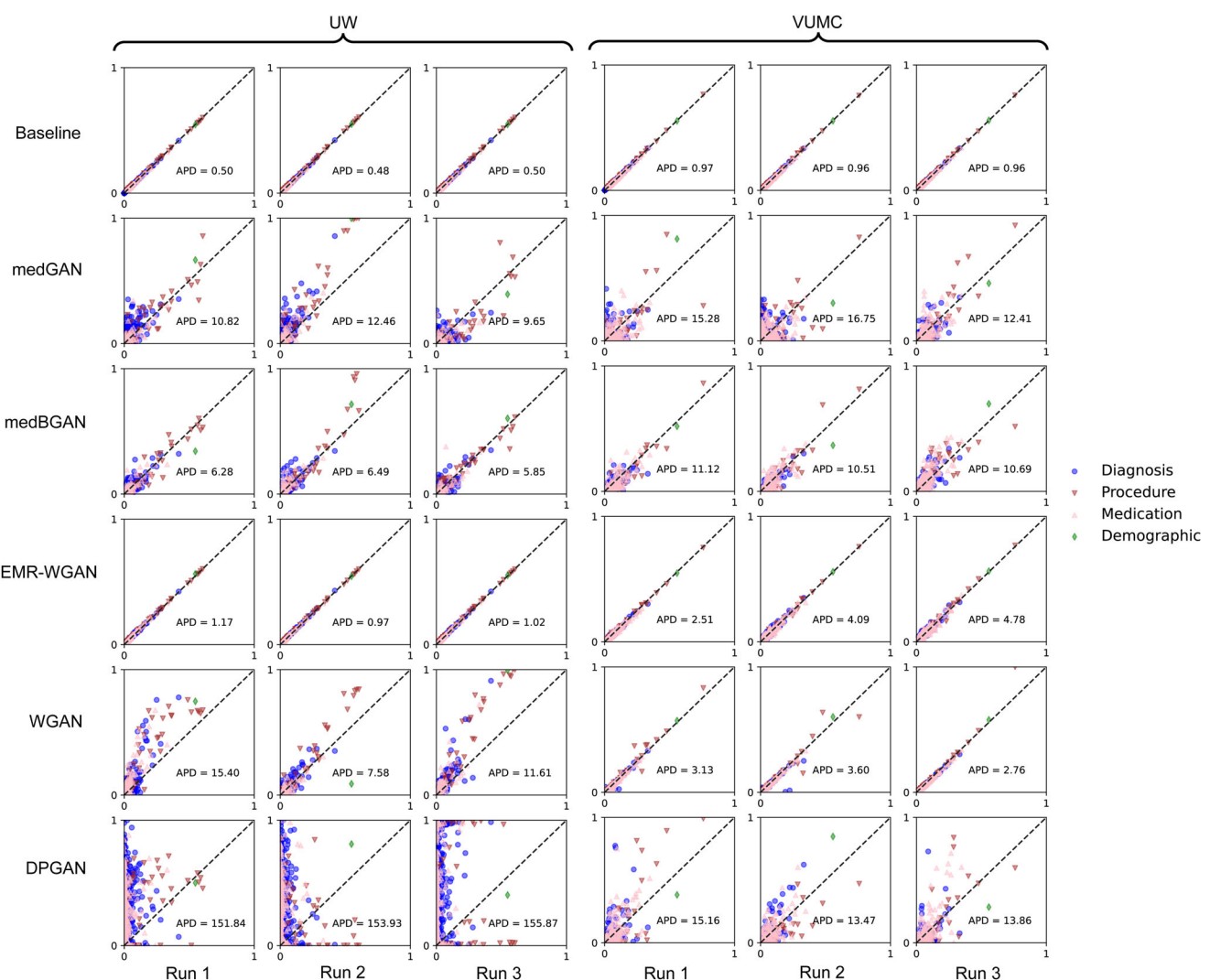

**Fig. 2 | Dimension-wise distribution for the UW and VUMC datasets.** Here, each dot corresponds to a feature, and the *x*- and *y*-axes correspond to the prevalence of a feature in real and synthetic data, respectively. The results for three independently generated synthetic datasets are shown for each candidate model. Dots on the dashed diagonal line correspond to the perfect replication of prevalence.

implies that the two metrics complement each other by discriminating model assessment. Figure 5c depicts a heatmap of the rank-derived scores (in Fig. 5a, b) across all models for the VUMC and UW datasets. There are several notable findings. First, the privacy-utility tradeoff is evident from the fact that all pairs of utility and privacy metrics were negatively correlated. Second, numerous weak correlations were observed in the utility metrics, including a very weak negative correlation between TRTS Model performance and dimension-wise distribution. In particular, the performance of synthesis models on dimension-wise distribution was weakly correlated with the model prediction performance, medical concept abundance, and clinical knowledge violation metrics. Yet, there were a relatively strong correlation (correlation coefficient = 0.89) between column-wise correlation and latent cluster analysis and two strong correlations for medical concept abundance with (1) latent cluster analysis (correlation coefficient = 0.90) and (2) TSTR model performance (correlation coefficient = 0.94). Third, from a privacy perspective, attribute inference risk exhibited weak correlations with the other three privacy metrics, whereas the membership inference risk rankings were correlated with meaningful identity disclosure risk in a relatively strong manner (correlation coefficient = 0.88). With only 2 (out of 66) metric pairs associated with a correlation coefficient >0.90, the results indicate that the metrics are complementary and belong in the framework.

## Model selection in the context of use cases

Generative model benchmarks need to be contextualized through specific use cases, where different uses of the synthetic data will have different priorities when it comes to the different metrics in the framework. In the Model recommendation phase, we apply three example use cases to illustrate the process and select the most appropriate synthetic data generation model: (1) education, (2) medical AI development, and (3) system development (as detailed in "Methods"). Each use case is associated with a different set of weights for the utility and privacy metrics. We adjusted the weights assigned to the ranking results from individual metrics for each use case according to their typical needs. For example, the utility weights for the first-moment statistics, medical concept correlations, patient-level clinical knowledge, and medical concept abundance were set higher in the education scenarios than in other use cases, whereas privacy risks were of less concern than in the system development and medical AI development use cases. By contrast, medical AI development provides greater emphasis on the metrics for model development and interpretability than the other two use cases. The system development use case places a greater emphasis on privacy risks and data sparsity to support function and data flow testing.

Table 4 summarizes the final ranking results for the generative models based on the use cases. It can be seen that EMR-

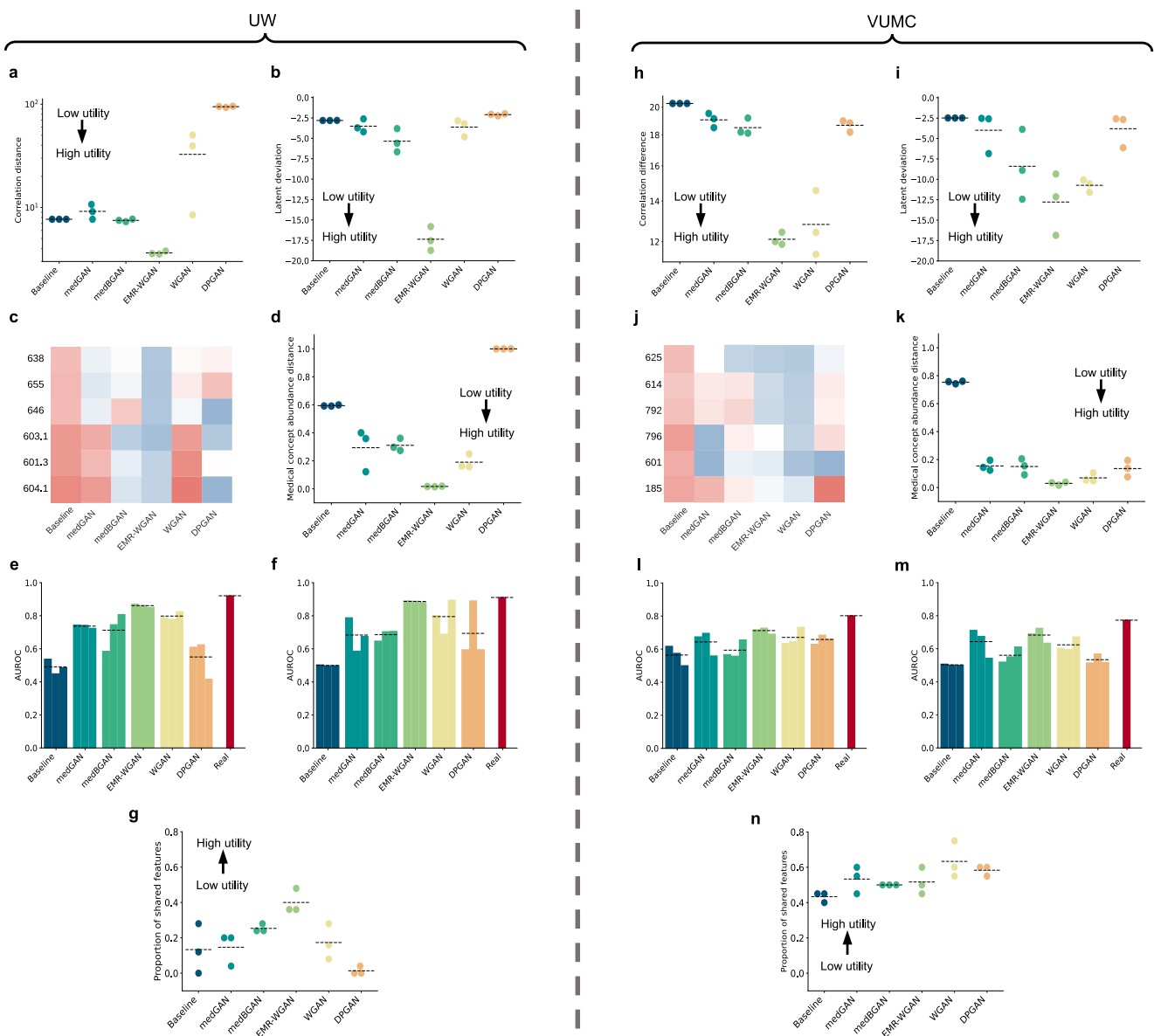

**Fig. 3 | Data utility for the UW (a–g) and VUMC (h–n) datasets. a, h** Column-wise correlation. **b, i** Latent cluster analysis. **c, j** Clinical knowledge violation for gender-specific phecodes. **d, k** Medical concept abundance. **e, l** TSTR Model performance. **f, m** TRTS Model performance. **g, n** The proportion of top *k* features in common (25 for UW and 20 for VUMC). The heatmaps correspond to the ratio of clinical knowledge violations in gender (blue = low value; red = high value). A dashed line indicates the mean value across three synthetic datasets. (Phecode: 625: Symptoms associated with female genital organs; 614: Inflammatory diseases of female pelvic organs; 792: Abnormal Papanicolaou smear of cervix and cervical HPV; 796: Elevated prostate-specific antigen; 601: Inflammatory diseases of prostate; 185: Prostate cancer; 638: Other high-risk pregnancy; 655: Known or suspected fetal abnormality; 646: Other complications of pregnancy NEC; 603.1: Hydrocele; 601.3: Orchitis and epididymitis; 604.1: Redundant prepuce and phimosis/BXO).

WGAN and WGAN best support education and medical AI development, respectively. By contrast, sampling-based Baseline was best for system development. It is notable that, in the system development use case, EMR-WGAN and DPGAN achieved the lowest rankings for the VUMC and UW, respectively. In addition, DPGAN demonstrated ranking scores that were no better than WGAN on all of the six scenarios considered (use case by dataset).

## Synthesis paradigms

A data synthesis paradigm is the manner by which a generative model utilizes features in a real dataset to generate synthetic data. These features could be a key outcome (e.g., a readmission event) or demographics (e.g., age of the patient). The selection of a synthesis paradigm has an impact on the utility and privacy of synthetic data. It should be noted that selecting synthesis paradigms and candidate

generative models are two facets of the synthesis process embedded into the Synthetic EHR data generation phase (Fig. 1). The benchmarking framework was thus designed to accommodate the need to incorporate different data synthesis paradigms according to the key features (e.g., the 21-day hospital admission post COVID-19 positive testing and six-month mortality in general) as part of the benchmarking. In this study, we specifically evaluated two synthesis paradigms: (1) the combined synthesis paradigm (Fig. 6c) and (2) the separated synthesis paradigm (Fig. 6d). Further details are provided in "Methods".

We observed that the separated synthesis paradigm demonstrated several advantages over the combined synthesis paradigm. The separated paradigm tended to achieve better utility on the dimension-wise distribution and outcome prediction metrics, while sustaining only a negligible increase of privacy risks (Supplementary Figs. 1–3,

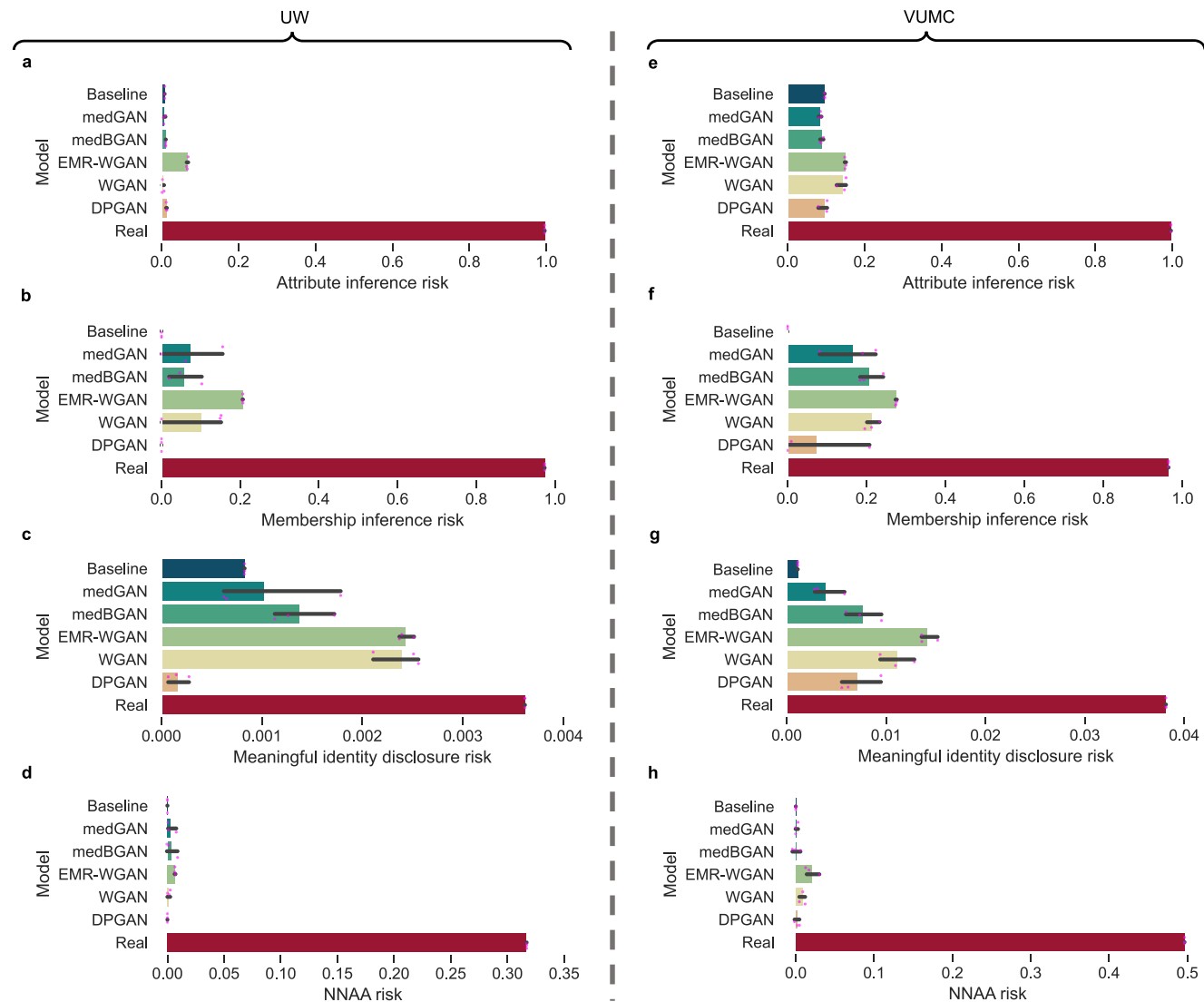

**Fig. 4 | Average privacy risk (*n*=3) of the synthetic versions of the UW (a–d) and VUMC (e–h) datasets. a, e** Attribute inference risk. **b, f** Membership inference risk. **c, g** Meaningful identity disclosure risk. **d, h** Nearest neighbor adversarial accuracy (NNAA) risk. The risks associated with the real data are shown in the bottom red bars. The 95% confidence intervals are marked as thin horizontal black lines.

Supplementary Table 3). In all use cases, the separated synthesis paradigm outperforms the combined synthesis paradigm for all six models (Supplementary Table 4), save for two situations in the educational use case, where the combined synthesis paradigm led to slightly better performance than the separated synthesis paradigm for medBGAN and WGAN.

The raw values for each evaluation metric are provided in Supplementary B.

## Discussion

The framework introduced in this work provides a mechanism to determine which EHR data synthesis models are most appropriate for which use case for a given dataset. The framework can further be applied to guide the development of new synthesis models by enabling greater consistency in evaluations. There are multiple aspects worth discussing.

The benchmarking results with the UW and VUMC datasets yielded several notable findings. First, EMR-WGAN consistently performs well for the majority of utility metrics (Fig. 5a, b and Supplementary Table 1). This benefit is clearly due to a sacrifice in privacy, as this method also consistently exhibited the greatest privacy risks. Second, the inverse is

true for Baseline, in that it achieves low privacy risks at the expense of low utility (except for dimension-wise distribution) (Fig. 5a, b and Supplementary Table 1). This is not surprising because its sampling strategy neglects the joint distribution of the real data. Third, DPGAN, which adds a small amount of noise to WGAN, achieved a similar or worse ranking than WGAN (Table 4). This implies that, at least for the settings considered in this study, there is not much benefit in incorporating differential privacy into the synthetic data generation process.

In addition, it is worth remarking that there are non-trivial differences in model performance between the UW and VUMC datasets. Notably, with respect to APD (Fig. 2), while WGAN and DPGAN perform well for the VUMC dataset, they do not for the UW dataset. This may stem from differences in the complexity of the joint distribution between the two real datasets and the WGAN mechanism applied to this dataset (note that DPGAN was implemented based on WGAN). At the same time, this finding may also be an artifact of the differences in the selected synthesis paradigms. It can be seen, for instance, that the separated synthesis paradigm led to an improved feature distribution resemblance (Supplementary Fig. 1).

Our analysis also provides evidence of a utility-privacy tradeoff across the synthesis models. This phenomenon has been widely

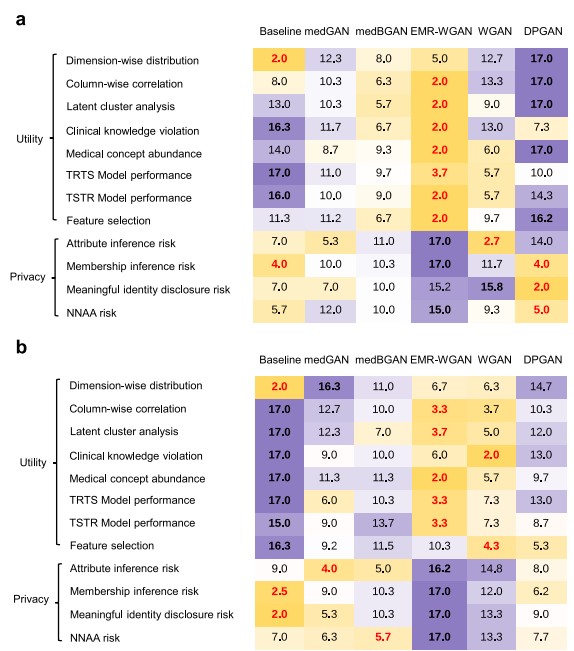

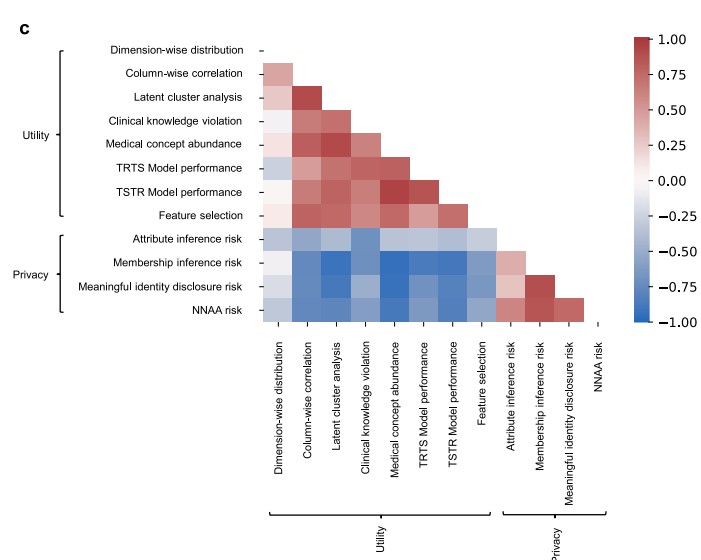

**Fig. 5 | Rank-derived scores of synthesis models and metric correlations.**
**a** Rank-derived scores on synthetic data generated using UW data. **b** Rank-derived scores on synthetic data generated using VUMC data. **c** A heatmap of the Pearson correlation coefficients for pairwise metrics on the rank-derived scores across all candidate models for the two real datasets (UW and VUMC). The best (i.e., lowest rank) and worst scores for each metric are in bold red and black font, respectively.

**Table 4 | Overall rank of generative models for the use cases in the Model recommendation phase**

| Use Case | Dataset | Overall model rank | | | | | |
|---|---|---|---|---|---|---|---|
| | | 1 | 2 | 3 | 4 | 5 | 6 |
| Education | UW | EMR-WGAN (5.6) | medBGAN (8.0) | Baseline (8.7) | medGAN (10.5) | WGAN (10.9) | DPGAN (13.2) |
| | VUMC | WGAN (6.5) | EMR-WGAN (7.5) | medBGAN (10.1) | Baseline (10.7) | DPGAN (11.0) | medGAN (11.2) |
| Medical AI Development | UW | EMR-WGAN (6.3) | WGAN (8.6) | medBGAN (8.7) | medGAN (9.9) | Baseline (11.2) | DPGAN (12.3) |
| | VUMC | WGAN (8.1) | DPGAN (8.5) | EMR-WGAN (8.6) | medGAN (8.8) | medBGAN (10.8) | Baseline (12.0) |
| System Development | UW | Baseline (7.9) | medBGAN (9.1) | EMR-WGAN (9.5) | medGAN (9.6) | WGAN (9.8) | DPGAN (11.1) |
| | VUMC | Baseline (8.7) | medBGAN (9.3) | medGAN (9.3) | WGAN (9.4) | DPGAN (9.5) | EMR-WGAN (10.8) |

Model ranks were based on the benchmarking framework scores (in parenthesis).
The fact that DPGAN and EMR-WGAN have the same score for the VUMC dataset in the Medical AI Development use case is due to precision loss instead of an actual tie.

discussed in the data privacy literature[42,43], where traditional privacy-preserving approaches (e.g., generalization or suppression of sensitive information) are applied to make changes directly to the real dataset that will be shared. Yet this phenomenon is characterized by several aspects of our study on synthetic data. First, no data synthesis model is unequivocally the best for all metrics, use cases, or datasets (Fig. 5a, b and Supplementary Table 1). Second, the overall ranks of data synthesis models differ across synthetic data use cases. For example, Baseline was among the worst models for Medical AI development, but was the best for System development (Table 4). Third, the evaluation metrics for data utility and privacy are negatively correlated in general (Fig. 5c). These findings highlight why it is critical to contextualize the comparison of EHR synthesis models through concrete use cases.

In investigating how evaluation metrics relate to each other, we observed two correlations that are stronger than others (Fig. 5). First, there was a relatively strong positive correlation between column-wise correlation and latent cluster analysis, which suggests that, if a GAN-based model retained correlations between features in real data, it was likely able to represent the joint distribution of the real data as well. This is because GANs do not explicitly learn from local feature correlations, but rather focus on global patterns. Second, medical concept abundance demonstrated a strong correlation with both latent cluster analysis and TSTR model performance. In effect, medical concept abundance serves as a proxy measure that characterizes how close the distributions of the severity of illness (or, more generally, health status) are between a synthesized cohort and the corresponding real cohort. For GAN models, medical concept abundance score that is close to zero implies that the synthesized cohort has high patient-level feature quality and thus can be clustered in a similar manner to real data in the latent feature space. These two aspects typically align with how TSTR model performance and latent cluster analysis work, respectively. Third, unsurprisingly, there was a relatively strong relationship between membership inference risk and identity disclosure risk. When a synthetic record has a high meaningful identity disclosure risk, it likely contains values that are very similar to a real record on a substantial portion of its sensitive attributes. Similarly, when a synthetic record has a high membership inference risk, there is a real record in the training data that it looks very similar to. Thus, it is likely that the two records match their quasi-identifiers and contribute to a high meaningful identity disclosure risk. However, these observations do not imply that there exists redundancy in metrics because (1) they measure clearly different data characteristics, (2) this observation might not generalize to other models and datasets that can lead to reduced correlations, and (3) the correlation coefficients are still deemed imperfect correlations.

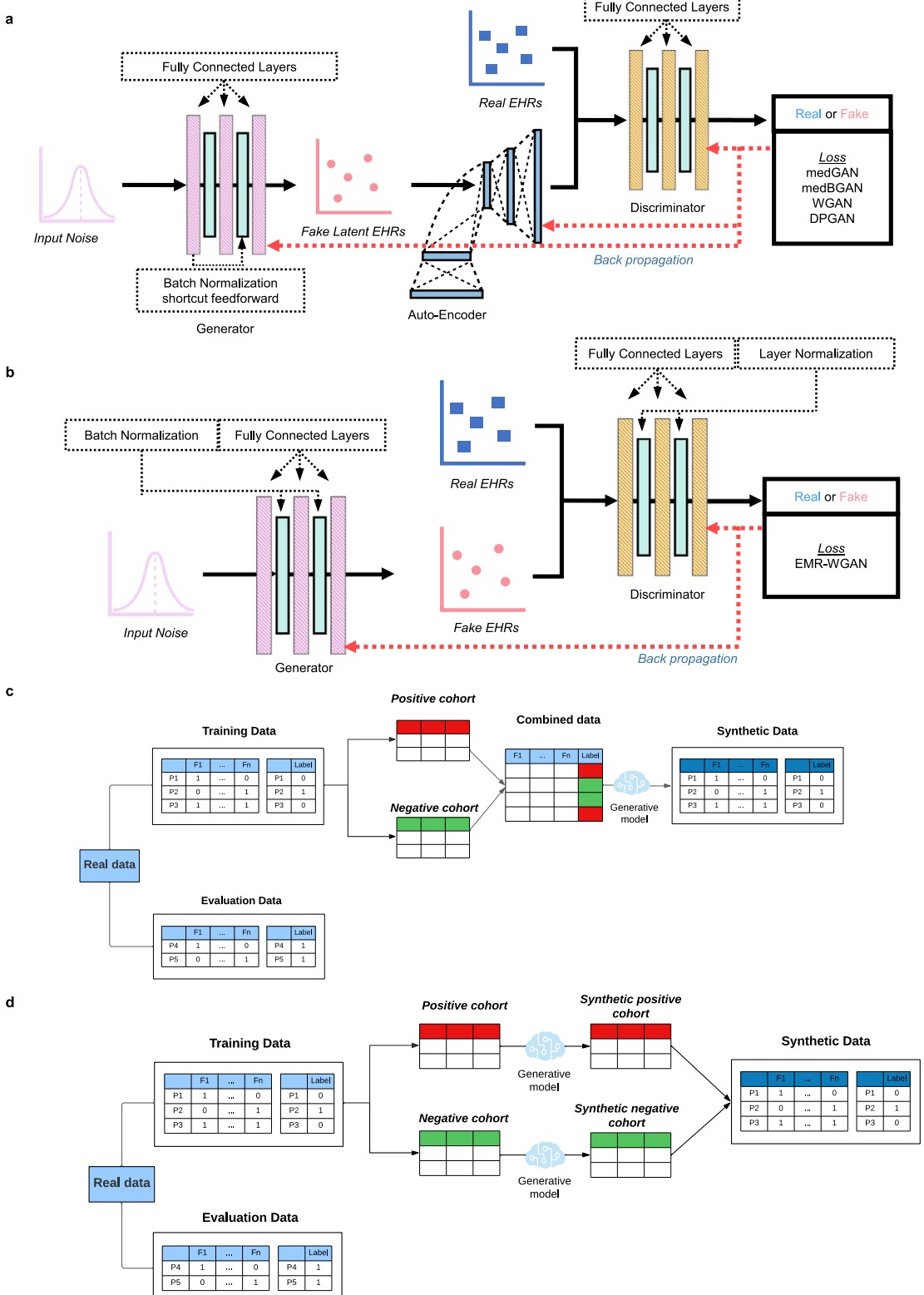

**Fig. 6 | An architectural depiction of the deep generative models and synthesis paradigms considered in this study. a** The generative model architecture of medGAN, medBGAN, WGAN, and DPGAN. **b** The generative model architecture of EMR-WGAN. **c** Combined synthesis paradigm. **d** Separated synthesis paradigm.

We observed that all models induced some level of knowledge violation, while the rates of violation at which the occurrence transpired differ between the methods. For instance, we observed a non-trivial number of violations in commonsense sex-disease relationships for all models (Fig. 3c, j). For instance, over 50% of the synthetic

records with the prostate cancer diagnosis code generated by DPGAN are associated with a generated Female gender. This phenomenon implies that GAN models are unable to perfectly recognize and learn from the record-level knowledge in EHR. Although such violations can be partially resolved through a post-hoc editing process before data

are shared, this phenomenon suggests that further research is necessary for the development of synthesis methods, such as embedding constraints of violations as a penalty into the learning process[21].

Despite the merits of this study, there are several limitations that provide opportunities for future improvement. First, the evaluation metrics we incorporated into the benchmarking framework do not necessarily represent the entire metric universe for synthetic data assessment. We aimed to cover the key characteristics of synthetic data by introducing representative metrics and did not incorporate every related metrics that could create redundancy. However, the framework can readily be extended to incorporate new utility and privacy metrics as they are introduced.

Second, this study relied upon a handful of use cases to assess the synthesis models and interpret the results that do not cover the gamut of all possible applications. As a result, the weight profiles applied to the metrics in the evaluation may not sufficiently represent the space. There is clearly an opportunity to explore how changing the weights influences model rankings. In particular, we neglected scenarios where the privacy risks were already sufficiently low, such that the synthetic datasets should only be assessed for their utility. Yet, the challenge in this scenario is that there is no clear consensus on what an acceptable privacy risk threshold for synthetic data is[29]. In this respect, we believe that further discussion and deliberation on legal standing and policy making is needed to inform the benchmarking framework[44]. On the other hand, when the goal of data synthesis is to support data augmentation[45,46] (rather than data sharing), where the limited volume and representativeness of the real data that are now available can be addressed to boost the effectiveness of medical AI algorithms[47–49], maintaining data utility becomes the primary goal of synthetic data generation. The benchmarking framework can provide support to data augmentation by ruling out privacy components by setting the weights of the related metrics to zero and incorporating new metrics, such as training on real and synthetic data testing on real data.

Third, the introduced benchmarking framework was specifically designed for contrasting models that generate structured EHR data. We believe that other data types in EHRs, such as longitudinal medical events, unstructured clinical notes, and medical images, are also valuable for data synthesis such that building corresponding benchmarking frameworks is critical as models emerge. The development of these frameworks requires incorporating new evaluation metrics that are specific to the nature of the data components. However, we believe our ranking strategy will still be reusable.

Fourth, we considered only a subset of GAN models for demonstration purposes. In doing so, we excluded models that were developed to resolve deficiencies in EHR synthesis that arise in specific applications. For instance, we did not include MC-medGAN, which allows for a better representation of multi-category features;[50] Cor-GAN, which was designed to represent correlations between physically adjacent features;[51] and HGAN, which considers constraints between features[21], and others that made minor adjustments to core GAN architecture.

Fifth, the two datasets utilized in our evaluation, though derived from real EHR systems, were curated through a series of preprocessing steps. As a result, they may not represent the full scope of EHR data complexity. For instance, they may not address heterogeneity in data modality, organizing structures, missingness patterns, and the size of feature space, among other aspects. As such, further investigation is needed to examine, and potentially extend, the applicability of the benchmarking framework for real-world datasets with various properties.

Sixth, there is mounting evidence that suggests generative models may induce bias and fairness issues, such that subgroups of the population are not evenly well-generated[52]. As a consequence, the resulting synthetic data, in certain circumstances, could accentuate disparities in health. We believe that the benchmarking framework introduced in this paper would benefit from extensions to cover bias and fairness dimensions. However, measuring bias and fairness in synthetic data is still very much an open problem and likely requires further investigation before a canonical set of metrics are ready for integration.

Seventh, the final results of the benchmarking framework are rank-based, which compresses the absolute values from individual metrics to relative values in the rank space. Thus, they are not directly comparable to the results in prior publications. A user might have to rerun all analyses on their own datasets in order to find the optimal model. Given this situation, there is clearly a need for universal metrics that can be compared across datasets for different models.

Eighth, the statistical significance of the observed differences for each metric cannot be tested in many cases due to the fact that there were only three runs of models. This would be possible if additional runs were performed, but each run required substantial computational resources such that there is no simple cost-effective way to feasibly do so at the present moment in time. However, our rank-based method does not rely on statistical significance to derive model comparison.

Finally, we aimed to use consistent parameterizations across the GAN implementations when they incorporated the same technical mechanism (e.g., the size of a deep network) while respecting the original implementations of all methods. However, the datasets we used in this study were not the same as those relied upon in the development of these methods. As a consequence, it is possible that the parameter settings we relied upon might not be optimal. Thus, a more extensive set of experiments is needed to investigate the generalizability of our observations on benchmarking results.

## Methods

The Institutional Review Boards (IRB) at Vanderbilt University Medical Center and the University of Washington approved this study under IRB#211997 and 00011204, respectively. The IRBs grant a full waiver of written informed consent from patients due to the nature of the retrospective observational study.

### Dataset

**University of Washington (UW).** This dataset comes from the general population at UW Medicine enterprise data warehouse, which manages EHR data from more than 60 medical sites across the UW Medicine system including the University of Washington Medical Center, Harborview Medical Center, and Northwest Hospital and Medical Center. Specifically, data from January 2007 to February 2019 for patients with at least 10 visits within two years prior to the date of the latest recorded visit were used. This dataset includes diagnoses, medications, procedures, and an indication of if the patient died within six months after the final visit date of the patient[53]. This dataset covers 188,743 patients.

**Vanderbilt University Medical Center (VUMC).** This dataset corresponds to a cohort of COVID-19 positive patients who visited VUMC. Specifically, we selected the patients who tested positive (via a polymerase chain reaction test) in an outpatient visit before February 2021. For those who exhibited multiple positive testing results, we retained one at random. We collected the diagnoses, medications, and procedures from these patients' EHRs between 2005 and the date of the positive COVID-19 test. In addition, the most recent readings for the seven most prevalent measures or laboratory tests prior to the selected positive testing events were included. This corresponded to diastolic and systolic blood pressures, pulse rate, temperature, pulse oximetry, respiration rate, and body mass index. For this dataset, the prediction task is whether a patient was admitted within 21 days of their COVID-19 test[54]. This dataset covers 20,499 patients.

To standardize data representation, we converted the categorical features, including diagnoses (encoded as International Classification of Diseases Ninth or Tenth Revision, or ICD9/10), procedures (encoded as Current Procedural Terminology Fourth Edition, or CPT4), medications (encoded as RxNorm Drug Terminology), and demographics (gender and race), to a binary format to denote the presence (or absence) of the corresponding concepts. We followed the convention of dimensionality reduction preprocessing[20,25,55] by (1) mapping the ICD9/10 codes into Phenome-wide Association Studies (PheWAS) codes, or phecodes, which aggregate billing codes into clinically meaningful phenotypes, (2) generalizing the CPT4 codes using a hierarchical architecture of procedures[56], and, (3) converting clinical RxNorm drugs to RxNorm drug ingredients. We represented the race of patients in a one-hot encoding format (i.e., a binary vector). We retained features with more than 20 occurrences in each dataset. After preprocessing, features in the UW dataset were all binary, whereas the VUMC dataset contained eight continuous features (namely, age and 7 laboratory tests).

Both datasets were split according to a 70:30 ratio into Training and Evaluation datasets. The Training datasets were relied upon to train synthetic data generation models while the Evaluation datasets were reserved for assessment purposes only.

## GAN models for benchmarking

Figure 6a, b illustrate the architectures for the GAN models assessed in the benchmarking activities.

medGAN was an early attempt to leverage the power of GANs[16] to synthesize individual-level EHR data[25]. The categorical format of medical concepts created a challenging situation in learning where the GANs' approximation of the discrete data rendered the training process suboptimal. To address this issue, medGAN leveraged a pre-trained autoencoder to project the discrete representations into a compact continuous space to enhance the subsequent GANs training. Also, medGAN integrated a set of helpful learning techniques, such as batch normalization and short connection, to reduce the instability of the training process. It should be recognized that medGAN relied upon the Jensen-Shannon Divergence (JS Divergence) between the real and synthetic data distributions as the optimization objective. These designs (except for the learning objective) were inherited by several later models as shown in Fig. 6a.

medBGAN was based on the same architectural designs as medGAN[35]. To enhance training performance in terms of the quality of the synthetic data, particularly when dealing with categorical values, medBGAN replaced the loss function of the medGAN discriminator with a boundary-seeking loss function. In doing so, the generator was directed to generate synthetic data points near the decision boundary of the discriminator, which enables better generation performance particularly for categorical data.

WGAN was developed based on the observation that a learning objective adopting JS divergence can lead to diminishing gradients, which, in turn, can subsequently impede the optimization of the generator[35]. To address this problem, WGAN incorporated Wasserstein divergence[57] into the training objective. This approach applies a Lipschitz constraint on the discriminator and ensures a more accurate characterization of the distance between two distributions. The WGAN implementation in our paper used the strategy that each of the discriminator's parameters is clipped to stay within a certain range (i.e., −0.01–0.01), which is referred to as parameter clipping, to satisfy the Lipschitz constraint.

DPGAN was a differentially private (DP) version of WGAN[36], which achieved a theoretically guaranteed privacy protection on synthetic health data via employing the differential privacy principle[58] in GAN training. It followed the differential private stochastic gradient descent (DP-SGD) mechanism in the model training process so that it is differential private, but modified the implementation of DP-SGD by replacing gradient clipping with parameter clipping. We set $\epsilon$ to $10^4$ to prevent the generation of synthetic datasets with very little to no utility.

EMR-WGAN was developed based on the observation that the autoencoder design can induce barriers during model training when working with an advanced distance measure between two distributions[20]. EMR-WGAN had the autoencoder component removed and equipped Wasserstein divergence as its optimization objective (Fig. 6b). EMR-WGAN made several additional amendments to the architecture. First, it introduced layer normalization to the discriminator to further improve the learning performance. Second, it incorporated a gradient penalty strategy to enforce the Lipschitz constraint, which reduced the negative impact of parameter clipping on GAN's capability to approximate data distribution.

## Model training and data selection

We normalized all continuous features by mapping them into a [0,1] range. We trained the generative models using the normalized data and mapped the generated data back to the original space of the continuous features as a post-training step. To enable a direct comparison, all hyperparameters were assigned the same values across all models in which they resided. For instance, we used the same deep neural network architecture, learning rates, optimizers, and initialization strategies.

Given that the GAN models differed in the pattern of training loss trajectories, to ensure a fair comparison such that the optimal training status for each model can be selected, we applied several rules to select the model training endpoint. We observed that the divergence loss for medGAN and medBGAN demonstrated a pattern of fluctuation before quickly growing to a very large number. We selected the epoch right before the beginning of the increasing trend, which usually corresponded to the lowest losses. By contrast, the losses of EMR-WGAN and WGAN demonstrated a clear convergence pattern, where the losses decreased first and then stayed relatively stable; however, the quality of synthetic data can differ after loss convergence. For both models, we examined multiple epochs from the area where the training loss converged and selected the top three synthetic datasets that demonstrated higher utility in dimension-wise distribution. For DPGAN, we observed that the loss decreased to a relatively low value and then started to fluctuate. We then selected the epoch right before fluctuation begins as the end point of training.

## Multifaceted assessment

**Data utility.** In earlier investigations[8,14], the term utility was defined in parallel with resemblance (i.e., the statistical similarity of two datasets) and was specifically used to refer to the value of real or synthetic data to support predictions. By contrast, in this work, we rely upon the terminology as it has been communicated in the literature[41,42,59,60], where privacy and utility are typically posed as competing concepts and utility indicates the general quantifiable benefit of data for its consumers. In this respect, the notion of resemblance is a specific realization of utility.

**Dimension-wise distribution.** The distributional distance of each feature between a real and synthetic dataset is often applied to measure data utility. In this study, we calculated the average of the absolute prevalence difference (APD) for binary features and the average of the feature-wise Wasserstein distances (AWD) for continuous features. The prevalence of a binary feature was defined as the percentage of patients who were associated with the corresponding concept in a dataset. Due to the fact that the Wasserstein distance is unbounded, for each continuous feature, we normalized the values into the range of [0,1] based on the distances derived from all synthetic datasets used for model assessment.

For a dataset with both binary (note that categorical features can be converted to binary features for synthesis) and continuous features, such as the VUMC dataset, the results of APD and AWD need to be combined into a final score. To do so, for binary features, we sum the absolute prevalence difference, and, for continuous features, we sum the feature-wise (normalized) Wasserstein distances. We then averaged the two results (i.e., divided by the total number of features). To ensure the average values are easy to read, we multiplied a factor of 1000 by the metric results, which share the same magnitude of the number of features.

**Column-wise correlation.** This metric quantifies the degree to which a synthetic dataset retains the feature correlations inherent in the real data[26]. For each pair of synthetic and real datasets, we first computed the Pearson correlation coefficients between all features in each dataset, which yielded two correlation matrices of the same size. We then calculated the average of all cell-wise absolute differences between the two matrices to quantify the fidelity loss in a synthetic dataset. We multiplied all values by a factor of $1000^2$ for presentation purposes. For reference convenience, we name this quantity as correlation distance.

**Latent cluster analysis.** This metric assesses the deviation of a synthetic dataset in the underlying latent space from the corresponding real dataset in terms of an unsupervised clustering[27]. For each pair of real and synthetic datasets, we stacked them into a larger dataset and reduced the dimensionality of the space by applying a principal component analysis (PCA) and retaining the dimensions that cover 80% of the variance in the system. We then applied $k$-means to define the clusters, where $k$ was determined according to the elbow method[61] (which was found to be three for both datasets in this study). It should be recognized that the elbow method is a heuristic and the number of clusters could alternatively be specified according to the desired granularity of quality inspection. The following clustering-based value was then calculated to quantify the deviation of synthetic data from real data:

$$\log\left(\frac{1}{K}\sum_{i=1}^{K}\left[\frac{n_i^R}{n_i}-0.5\right]^2\right) \qquad (1)$$

where $n_i^R$ and $n_i$ denote the number of real data points and the total number of data points in the $i^{th}$ cluster, respectively. We name this quantity as latent deviation for reference convenience. For this metric, a lower value implies that the density functions of the real and synthetic datasets in the latent space are more similar.

**Clinical knowledge violation.** Unlike the previous metrics, this metric focuses on the record-level utility. Specifically, it quantifies the degree to which a generative model learns to synthesize clinically meaningful records in terms of the ability to capture clinical knowledge from data. An example for continuous features is that the systolic blood pressure of a patient should be greater than the corresponding diastolic pressure. A synthetic dataset with a number of conflicts against clinical knowledge derived from real data can be less useful in those use cases that rely on record-level readability. In this study, we used a data-driven approach to derive the clinical knowledge from real data for model evaluation. We first identified the phecodes from real data that were only associated with one gender. For each gender, we then selected the most prevalent three phecodes that were only associated with this gender in real data. For each synthetic dataset, we computed the odds of each selected phecode appearing in the opposite gender. The average value from the selected phecodes was then calculated, where a higher value implies a lower capability of representing the clinical knowledge inherent in the real data.

**Medical concept abundance.** Close resemblance in feature-level metrics does not necessarily imply high similarity in the record-level distributions between the real and synthetic datasets. Thus, inspired by the work of Yale et al.[30], we introduce a metric that characterizes the degree to which a synthesis model captures the quantity of record-level information in the real data. Specifically, we compute the normalized Manhattan distance between the histogram of the number of assigned distinct medical concepts for real and synthetic records. To do so, we fix the space to the total number of distinct medical concepts that are considered in synthesis and then divide the range into $M$ evenly sized bins according to the desired assessment granularity. We then computed the medical concept abundance distance as $\sum_{i=1}^{M}|h_r(i)-h_s(i)|/2N$, where $h_r(i)$ and $h_s(i)$ represent the number of records in the $i^{th}$ bin that are real and synthetic, respectively, and $N$ denotes the total number of real records. This metric is thus in the [0,1] range, where a lower the value indicates a higher real-synthetic data similarity in terms of record-level information distributions. In this study, we set $M$ equal to 20.

**Prediction Performance and important features.** One of the more common scenarios for which synthetic EHR data is expected to provide support is machine learning model development and evaluation[62,63]. To assess this capability, we performed two types of analysis. The first, which is straightforward and has been widely utilized, compares model performance for a specific prediction task in two distinct scenarios: (1) training a machine learning model using the synthetic dataset (obtained from a generative model learned from a real dataset) and then perform an evaluation based on an independent real dataset, and (2) training a model based on the independent real dataset and evaluate it using the synthetic dataset. In each scenario, for comparison purposes, the reference model is trained based on the corresponding real dataset. The first scenario adheres to how the synthetic data will be utilized after data are shared in practice. By contrast, the second plays a complementary role in that it assesses how convincingly the synthetic records match their labels[28]. It should be noted that, in the second scenario, it is possible that the testing performance for certain synthetic data is higher than for the real data because of the potential of mode collapse for GAN models (which means a generative model can only generate synthetic records that are close to a subset of real data). However, we do not believe this is a concern, due to the fact that the other utility metrics will reflect this problem in the final model ranking.

For prediction, we applied light gradient boosting machines (LightGBMs) due to their consistently superior performance over traditional machine learning models in healthcare[64–66]. In this evaluation, we randomly partitioned each real dataset according to a 70:30 split, where the 30% data served as the independent real dataset. We use the area under the receiver operating characteristic curve (AUROC) as the performance measure. We used bootstrapping to derive a 95% confidence interval for each model.

The second analysis focuses on the degree to which a synthetic dataset provides reliable insights into important features in the prediction task. This was incorporated as a critical metric because model explainability is critical for engendering trust and conducting algorithmic audits. To do so, we counted the shared top important features for models trained on a synthetic dataset and the corresponding real dataset and then computed the corresponding proportion. We used the SHapley Additive exPlanations (SHAP)[67] value to rank features and defined the important features as the top $M$ features that retain 90% of the performance on real data, which was 25 and 20 for the UW and VUMC datasets, respectively.

## Privacy
We focused on three types of privacy attacks that have targeted fully synthetic patient datasets: attribute inference[20], membership

                                                                                          

inference[20,31], and meaningful identity disclosure[29]. In an attribute inference attack, given the synthetic dataset and partial information (e.g., demographics and phenotypic attributes) of a patient's record in the real dataset, an adversary can infer all sensitive attributes of the record. Henceforth, we use real dataset to denote real training dataset for simplicity. In a membership inference attack, given the synthetic dataset and a patient's record, an adversary can infer whether the patient's record is in the real dataset, which discloses sensitive information shared by all records in the real dataset (e.g., an HIV-positive dataset). In a meaningful identity disclosure attack, given the synthetic dataset and a population dataset with identifiers, an adversary can infer the identity (and sensitive attributes) of a patient's record in the real dataset by matching an identified record to a record in the synthetic dataset which also matches a record in the real dataset due to potential overfitting of the data generation process. In general, among the three attacks, a larger number of attributes (i.e., identities and sensitive attributes) can be inferred for each victim in the meaningful identity disclosure attack, whereas a smaller number of attributes (i.e., the membership) can be inferred for each victim in the membership inference attack. In addition, we harnessed a privacy loss metric, proposed by Yale et al.[30], which directly measures the extent to which a generative model overfits the real dataset.

**Attribute inference risk.** In an attribute inference attack[20], an adversary attempts to infer a set of sensitive attributes of a targeted record in the real dataset given a set of the targeted record's attributes and the synthetic dataset. The set of attributes known by the adversary usually includes demographic attributes, such as age, gender, or race[68]. Sometimes, the adversary also knows the target's common clinical phenomena such as a diagnosis of the flu, cold, stomach ache, or conjunctivitis. In these cases, the sensitive attributes correspond to the target's other diseases. We assume the adversary attempts to infer the attributes using a $k$-nearest neighbors (KNN) algorithm. More specifically, the adversary first finds the set of $k$ records in the synthetic dataset that are the most similar to the targeted record based on the set of known attributes as the neighbors. Given this set, the adversary attempts to infer each unknown attribute using a majority rule classifier for the members in the set.

To evaluate the attribute inference risk, we first calculated the inference risk for each attribute that the adversary wants to infer for a set of patient records. For each binary attribute, we simulated the inference attack and calculated the F1 score. Each categorical attribute was converted into binary attributes using one-hot coding. For each continuous attribute, we simulated the inference attack and calculated the accuracy, which is defined as the rate that the prediction is sufficiently close to the true value according to a closeness threshold. Afterward, we set the attribute inference risk measure as a weighted sum of the risks for attributes, where the weight for each attribute is proportional to the corresponding information entropy in the real dataset and all weights sum to one.

We used the entire real dataset as the set of targeted records. In the KNN algorithm, we set $k$ to 1 and use the Euclidean distance measure. We assumed that the adversary knows the demographic attributes (age, gender, and race for the VUMC dataset; gender, and race for the UW dataset) and the 256 phecodes that are most frequent in the real dataset. The adversary attempts to infer all of the other phecodes and numerical attributes. In this study, the closeness threshold was set to 0.1. In two supplementary experiments, we varied the setting by changing $k$ to 10 or changing the number of known phecodes to 1024.

**Membership inference risk.** Knowing that an individual corresponds to a record in the real dataset constitutes a privacy risk because the records may be included according to specific criteria. For instance, these criteria may be disease- (e.g., HIV) or lifestyle-dependent (e.g., a certain sexual orientation). Notably, this information may not be

included as an attribute in the real dataset because it is shared by all records in the dataset (e.g., when all of the real records select for their HIV positive status) and, thus, cannot be inferred in the aforementioned attribute inference attack. The adversary, with the knowledge of all or partial attributes of a target, can infer the membership by comparing the targeted record to all records in the synthetic dataset on those known attributes. A correct inference would reveal the target's sensitive information and also discredit the data sharer who aimed not to reveal that a certain individual was in the training data.

To evaluate the membership inference risk[20], we assume that the adversary is in possession of the synthetic data and all attributes of a set of targeted records. We further assume that risk evaluators (i.e., users of the benchmarking framework) know whether each targeted record is in the real dataset. We first calculate the Euclidean distance between each synthetic record and each targeted record in terms of all attributes. Given a distance threshold, the adversary claims that a targeted record is in the real dataset if there exists at least one record with a distance smaller than the threshold. After the adversary infers the membership status of the targeted records, the F1 score of the membership inference would be used as the risk measure.

We use all records in the real dataset and the evaluation dataset as the targeted records. We normalize all continuous attributes into a range of zero and one. We set the distance threshold to 2 in the main experiment and 5 in the supplementary experiment to assess the sensitivity of the model.

**Meaningful identity disclosure risk.** Although a fully synthetic dataset appears to have no risk of identity disclosure, a synthetic dataset generated by an overfitted machine learning model may permit record linkage to the original records. In recognition of this fact, El Emam et al. introduced a risk model[29] that considers both identity disclosure and the ability of an adversary to learn new information upon doing so. In this attack, the adversary links the synthetic dataset to a population dataset, which is an identified dataset that covers the underlying population of the real dataset, upon quasi-identifiers (i.e., the common attributes in both datasets). Afterward, for each targeted record in the synthetic dataset, the adversary infers the identity using a majority classifier over the linked records in the population dataset. They further assumed that the adversary can execute the record linkage attack by generalizing any attribute in any record to a certain level (i.e., an age, 20, can be generalized to an age group, [20–29]).

To evaluate the meaningful identity disclosure risk[29], we use the metric introduced by El Eman et al., which is based on the marketer risk measure and additionally considers the uncertainty and errors in the adversary's inference. This metric is calculated as:

$$\max\left(\frac{1}{N}\sum_{s=1}^{n}\left(\frac{1}{f_s}\times\frac{1+\lambda_s}{2}\times I_s\times R_s\right),\frac{1}{n}\sum_{s=1}^{n}\left(\frac{1}{F_s}\times\frac{1+\lambda_s}{2}\times I_s\times R_s\right)\right) \quad (2)$$

where $N$ is the number of records in the population, $s$ is the index for a record in the real dataset. $n$ is the number of records in the real dataset, $f_s$ is the number of records in the real dataset that can match record $s$ in the real dataset in terms of values on the quasi-identifiers (QIDs), $F_s$ is the number of records in the population that can match record $s$ in the real dataset in terms of values on the QIDs, $\lambda_s$ is an adjustment factor based on error rates sampled from two triangular distributions[29], $I_s$ is a binary indicator of whether record $s$ in the real dataset matches a record in the synthetic dataset, $R_s$ is a binary indicator of whether the adversary would learn something new, and $R_s$ is 1 if at least $L$% of the sensitive attributes satisfy the following criteria. For each categorical attribute, the criteria are: (1) there is at least one synthetic record that can match at least one real record on that sensitive attribute, and (2) $p_j < 0.5$ in which $p_j$ is the proportion in the real sample that have the same $j$ value, and $j \in J$ in which $J$ is the set of different values the sensitive feature can take. For each continuous attribute, the criterion

is $p_s \times |X_s - Y_t| < 1.48 \times MAD$, in which $p_s$ is the proportion in the real sample that are in the same cluster with the real record after a univariate $k$-means clustering, $X_s$ is the sensitive attribute of the real record, $Y_t$ is the sensitive attribute of the synthetic record matching the real record, and $MAD$ is the median absolute deviation.

In this study, we rely upon an adversarial model that is as strong as the one introduced by El Eman et al.[29] (i.e., with a similar sample-to-population ratio and a similar number of QIDs). For the VUMC dataset, we assume that the adversary has access to a population dataset of 633,035 records, which includes the name and 10 QIDs of all patients that have visited VUMC before February 2021. The corresponding QIDs are three demographic attributes (namely, age, sex, and race) and seven phenotypic attributes that are the most frequent diseases of those patients. For the UW dataset, we assume that the adversary has access to a population dataset of 466,980 records including the name and 10 QIDs of all patients who have visited UW at least five times in the past two years prior to the index event date, which is defined as the date of the latest recorded visit as of February 2019. The corresponding QIDs are two demographic attributes (namely, sex and race) and eight phenotypic attributes that are the most frequent diseases of those patients. The parameter $L$ is set to 1 which means at least 26 (or 27) attributes need to be inferred correctly and meaningfully for an attack to be regarded as a successful attack that brings risk to VUMC (or UW) dataset. The difference between VUMC and UW datasets is due to the fact that the VUMC dataset has 74 fewer attributes than the UW dataset. To test the sensitivity of the model, we change the parameter setup by changing $L$ to 0.1 in the supplementary experiment.

**Nearest neighbor adversarial accuracy risk.** Overfitting can induce privacy risks for synthetic data. Yale et al. introduced a privacy loss metric for synthetic data that directly measures the extent to which a generative model overfits the real dataset based on the notion of nearest neighbor adversarial accuracy (NNAA)[30,69]. The NNAA risk we use in our evaluation is based on this metric. Specifically, let $S_T$, $S_S$, and $S_E$ be three sets of samples with the same size from three datasets: $S_T = \{x_T^1, \cdots, x_T^n\}$ from the real training dataset, $S_S = \{x_S^1, \cdots, x_S^n\}$ from the synthetic dataset, and $S_E = \{x_E^1, \cdots, x_E^n\}$ from the evaluation dataset. The NNAA risk is calculated as the difference between two distances:

$$AA_{ES} - AA_{TS}$$

in which

$$AA_{ES} = \frac{1}{2}\left(\frac{1}{n}\sum_{i=1}^{n}1(d_{ES}(i) > d_{EE}(i)) + \frac{1}{n}\sum_{i=1}^{n}1(d_{SE}(i) > d_{SS}(i))\right) \quad (3)$$

$$AA_{TS} = \frac{1}{2}\left(\frac{1}{n}\sum_{i=1}^{n}1(d_{TS}(i) > d_{TT}(i)) + \frac{1}{n}\sum_{i=1}^{n}1(d_{ST}(i) > d_{SS}(i))\right) \quad (4)$$

where the indicator function $1(\cdot)$ equals one if its argument is true and zero otherwise. $d_{TS}(i) = \min_j \|x_T^i - x_S^j\|$ is defined as the distance between $x_T^i \in S_T$, a data point in the sample from the real data, and its nearest neighbor in $S_S$, the sample from the synthetic data. Furthermore, $d_{ST}(i) = \min_j\|x_S^i - x_T^j\|$ is the distance between $x_S^i \in S_S$, a data point in the sample from the synthetic data, and its nearest neighbor in $S_T$, the sample from the real data. Moreover, $d_{TT}(i) = \min_{j,j\neq i}\|x_T^i - x_T^j\|$ is the distance between $x_T^i$ and its nearest neighbor in a sample of size $(n-1)$ instances drawn from the same distribution, and $d_{SS}(i) = \min_{j,j\neq i}\|x_S^i - x_S^j\|$ is the distance between $x_S^i$ and its nearest neighbor in a sample of size $(n-1)$ instances drawn from the same distribution. Similarly, $d_{ES}(i) = \min_j\|x_E^i - x_S^j\|$ is the distance between $x_E^i \in S_E$, a data point in the sample from the evaluation data, and its nearest neighbor in $S_S$, the sample from the synthetic data.

Furthernore, $d_{SE}(i) = \min_j\|x_S^i - x_E^j\|$ is the distance between $x_S^i$, a data point in the sample from the synthetic data, and its nearest neighbor in $S_E$, the sample from the evaluation data. Moreover, $d_{EE}(i) = \min_{j,j\neq i}\|x_E^i - x_E^j\|$ is the distance between $x_E^i$ and its nearest neighbor in a sample of size $(n-1)$ drawn from the same distribution.

The original NNAA risk as defined by Yale et al.[30] requires all samples of datasets to have the same size; however, in our experiments, the sizes of the training and synthetic dataset are both larger than the size of the evaluation dataset. Thus, we randomly sample the training dataset and the synthetic dataset to be in the same size of the evaluation dataset multiple times and use the average result as the NNAA risk. In addition, we normalize all continuous attributes into a [0, 1] range before computing distances.

### Ranking mechanism
Supplementary Fig. 4 provides a concrete example of how our benchmarking framework ranks models. In this example, we use three candidate synthesis models to illustrate the process. When ranking models, with respect to each metric, each model receives a rank-derived score that is calculated as the average of ranks of three datasets associated with each model. The final score for each model is calculated as the weighted sum of the rank-derived scores from all metrics. All models are then ranked according to their final scores.

When ranking datasets, ties (i.e., two or more datasets having exactly the same value of a metric) can occur. In this case, the datasets receive the same adjusted rank (which is not necessarily an integer). In the example, there are three datasets that are tied in terms of an evaluation metric (Metric 2). The associated indices are 3, 4, and 5. The average of these three indices is thus $(3 + 4 + 5)/3 = 4$, which is taken as the adjusted rank that each of the three datasets would be assigned.

### Use case description
In this study, we consider three use cases of synthetic data to demonstrate generative model selections in the context of specific needs. The benchmarking framework translates a use case into weights on the metric-level results. By default, all weights are set to be equal, and all weights sum to 1. We adjusted the weights according to the needs of the use case. The following provides a summary of the use case, while the detailed weight profiles are provided in Supplementary Table 23.

**Education.** It is expected that synthetic EHR data will support educational purposes. The potential data users for this use case are students interested in health informatics or entry-level health data analytics. In general, privacy risks in this use case are relatively small for several reasons: (1) access control and audit logs are easy to implement, and (2) data use agreement can be applied to further protect the data. By contrast, the educational use case has high demand in maintaining the statistical characteristics of the real medical records and minimizing obvious clinical inconsistencies. Thus, we lowered the weight assigned to each of the privacy metrics to 0.05 and raised the weight for dimensional-wise distribution to 0.25, column-wise correlation, and clinical knowledge violation to 0.15, medical concept abundance to 0.10, with the remaining metrics set to 0.05.

**Medical AI development.** It has been repeatedly shown that synthetic health data are able to support the development of medical AI by providing similar testing performance as on real data[10,70]. We established the use case for machine learning model development based on the aforementioned prediction tasks: (1) 21-day hospital admission post positive COVID-19 testing (for the VUMC dataset) and (2) six-month mortality (for the UW dataset). In this use case, we prioritized the model prediction-related utility metrics (performance results and feature selection) and privacy because the synthetic data are open to

the broad data science community. As such, we raised the combined privacy weight to 0.3 and model performance to 0.5.

**System development.** In the healthcare domain, software system development teams often need access to sufficiently large and realistic datasets that mimic real data for function and workflow testing, as well as computational resource estimation. In this use case, it is important that the synthetic data maintain both the size and the sparsity of the real data. These are factors represented in metrics for dimension-wise distribution and medical concept abundance. At the same time, privacy needs to be prioritized as the engineers may not have the right to work with the records of patients—particularly if they are not employees of the healthcare organization. Thus, we set the dimension-wide distribution weight and medical concept abundance each to 0.15, the privacy metrics to 0.5, while each of the rest weights were set to 0.04.

### Synthesis paradigms

In this study, we investigated two common synthesis paradigms as examples. The first strategy treats the outcome variable the same as other features in model training, which leads to a combined synthesis paradigm (Fig. 6c), whereas the second strategy was designed to independently train a generative model for each outcome represented by the outcome variable, leading to a separated synthesis paradigm (Fig. 6d).

We applied the combined synthesis paradigm to both datasets and all results that have been communicated so far were based on this strategy. We performed a comparison between the two strategies on the UW dataset. We did not conduct the comparison for the VUMC dataset because the volume of positive records is too small to support separated GAN training. We ensured that the synthesized data shared the same size as the corresponding real dataset and that the distribution of the outcome variable remained the same as well.

More specifically, for each metric in the Multifaceted assessment phase, the number of synthetic datasets for evaluation becomes $n_m \times n_d \times n_s$, where $n_m$, $n_d$, and $n_s$ denote the number of candidate generative models for benchmarking, the number of synthetic datasets considered for each model in comparison, and the number of considered synthesis paradigms, respectively. In this investigation, we have $6 \times 3 \times 2 = 36$ synthetic datasets for the UW dataset.

### Reporting summary

Further information on research design is available in the Nature Portfolio Reporting Summary linked to this article.

## Data availability

The electronic health record data that support the findings of this study are available upon request from the corresponding authors and approval from the institutions' respective IRBs. Requests for access will be processed within around 2 months subject to signing of a data use agreement.

## Code availability

The source code associated with this study is publicly available at: https://github.com/yy6linda/synthetic-ehr-benchmarking.

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

## Acknowledgements
C.Y. was supported by NIH grants OT2OD032581 and U2COD023196. Z.W. was supported by NIH grant RM1HG009034. S.M. was partially supported by NIH grants UL1TR002319, P30AR072572, and P50AG005136. B.A.M. was partially supported by NIH grants OT2OD032581 and U54HG012510. This research was also supported by NIH grant UL1TR002243.

## Author contributions
C.Y., Y.Y., and Z.W. contributed to the conception and design of the study. C.Y., Y.Y., Z.W., and Z.Z. conducted the analysis and interpreted the results. C.Y., Y.Y., and Z.W. led the writing of the manuscript and are co-first authors. B.A.M., L.O., J.G., and S.M. revised the manuscript. B.A.M. and S.M. jointly supervised the research. All authors helped shape the research, analysis, and manuscript. All authors wrote the manuscript. All authors read and approved the final manuscript.

## Competing interests
The authors declare no competing interests.
