## [Peer Review File · Nature Communications]

Reviewers' Comments:

Reviewer #1:

Remarks to the Author:

In the abstract the work proposes a systematic benchmark framework to assess utility and privacy of tabular synthetic health data. "Privacy" is the extent to which patient data privacy is maintained. "Utility" is how well the synthetic works for the intended use such as a classification problem. There is normally a third assessment area which is "Resemblance" – which is how close is the synthetic data to the real data. The authors chose to roll together utility and resemblance into just utility. So, the paper does assess resemblance even if the abstract doesn't say it. The paper would be clarified and strengthened if they correctly broke out these three categories. It is important to do so. A single dataset can commonly be used for many purposes thus many different utilities could be assessed. A dataset can have poor resemblance but still do a good job for a specific task. If resemblance was perfect, then utility would be good for most any task. For the purpose of this review, we will consider resemblance to be different than utility and we strongly recommend that future revisions of the paper do this as well.

The paper addresses an important issue (benchmarking methods) in an important emerging area (synthetic health data). The authors do a good job of saying what it is important. They address how to systematically compare a bunch of different synthetic data generation methods. The many metrics used to inform the rankings are primarily existing metrics that have been used in the literature. They develop a three-phase approach which produces individual metric evaluations and then a ranking of the methods for that data set and task. The methods they proposed are well described and reasonable for deciding the best approaches for a given synthetic data problem. A weakness is the final metrics are ranked-based so the results from one paper can't easily be compared to the results in other papers. To show my synthetic method is better than prior methods using this benchmark, I have to rerun from scratch all the entire methods. Many of the metrics are not scaled, so it is not easy to know if the score is good or bad between different datasets. So, while it is certainly very useful and important to identify top models for a specific problem and their approach is clever, there is still a lot of room for future work in practical synthetic data benchmarking. In the end the topped ranked methods might not be that good.

The approach is based on a suite of data utility, resemblance, and privacy metrics. They did a nice job on the task specific utility metrics which include model performance and feature selection metrics. The measure of shared features is very algorithm specific and dependent on the total number of features. Other metrics might be stronger than a difference in the count – perhaps you could borrow NLP metrics for comparing two search results? Then one could tell if the results were strong or not.

They have privacy metrics: membership inference, meaningful identify disclosure, and attribute inference. They missed the nearest neighbor adversarial accuracy metric which was designed for specifically for assessing privacy in synthetic data (see Yale 2020 papers).

It's great that they included a clinical knowledge violation metrics. When available this is a great thing to do. But what about all the other inaccurate relationships that the synthetic data introduce? You need more general resemblance metrics that will capture this. One possibility is some metric based on sums of phecodes per patient. MEDGAN was found to generate unrealistic patients with many diagnosis codes in Yale 2020.

The other choices of resemblance metrics seem necessary but not sufficient. Mode collapse is a huge problem with GANs so subgroups of the dataset may be lost. GANs are tricky to train. In Figure 2, DPGAN is clearly failing catastrophically. The univariate marginal metric: dimension wise distribution metric, is okay – but taking average or Wasserstein distances on very high dimensional can mask the fact that rare data classes distribution are altered from real to synthetic data. The remaining multivariate metrics seem lacking. Column wise correlation only captures simple linear correlations. The Latent cluster analysis was a good idea but, in the end, only three clusters were used. This could easily miss mode collapse. Some other metrics that can make sure that all the training data has a reasonable synthetic counterpart (but not too close) and that vice versa are needed sorely needed. Recent studies have shown that synthetic data can introduce

significant biases when resemblance and fairness is evaluated with respect to critical covariates (Bhanot 2021). Would these measurements capture these types of issues with synthetic data? How could you make them stronger?

They evaluated on two datasets which seemed like good choices.

It's great that they allowed different weighting of metrics for different scenarios.

Are the observed differences significant? Could you incorporate statistical significance?

In Figure 2, DPGAN and WGAN on UW, it looks like DPGAN and WGAN are taking very rare features and making them very common. This shows a catastrophic failure of these two methods and is a bit unusual for a GAN result. Could you improve tuning? Also, it is possible that y is real and x is synthetic in Figure 2? Then this would be consistent with a mode collapse which is quite common. Probably worth a comment on why the methods are doing so poorly. Also, you used a huge value of epsilon so no theoretical proofs of privacy here.

The section on synthetic paradigms is not very clear. Synthetic paradigms were not defined earlier in the paper. It only works on one of the datasets. The results are primarily in the supplement. The topic seems kind of gratuitous and not the main subject of the paper. The contribution is not clear. The paper is about benchmarks not better methods. Recommend keeping for a later paper. Hopefully, your new improved benchmarks will stimulate lots of better reproducible synthetic data generation methods in the future.

For membership inference risk, the description could be clearer. "we assume that the adversary is in possession of all attributes of a set of targeted records and we know whether each targeted record is in the real dataset" is not clear. Who is we? Does this include the adversary? Does the attacker have the full set of synthetic data as well. Does the attacker leverage the synthetic data? Shouldn't you evaluate the synthetic data as well? The work on Nearest Adversarial Accuracy could be useful here.

Yale A., Dash S., Dutta R., Guyon I., Pavao A., Bennett K.P. Generation and evaluation of privacy preserving synthetic health data. *Neurocomputing*. 2020;416:244–255.

Yale A., Dash S., Bhanot K., Guyon I., Erickson J.S., Bennett K.P. Synthesizing Quality Open Data Assets from Private Health Research Studies. In: Abramowicz W., Klein G., editors. *Business Information Systems Workshops*. Springer International Publishing; Berlin, Germany: 2020

Bhanot, K. et al. The Problem of Fairness in Synthetic Healthcare Data, *Entropy* 2021.

Reviewer #2:

Remarks to the Author:

This is an excellent and well-timed manuscript describing a comprehensive framework for the evaluation of generative synthetic data. The authors are well-recognized experts, the paper employs rigorous methods, and the conclusions are theoretically and empirically strong. Further, the methods used should serve as the basis for reproducible methods in this domain. In many ways, the work described in the paper is overdue relative to the proliferation of synthetic data generation methods. As such, this reviewer only has a few minor suggestions, as follows:

- The summary of generative methods is rather cursory. While this is not the core premise of the paper, it is important, and many readers may not be familiar with such approaches. One possible solution to this would be the addition of a table or other construct that could summarize current (recommended) literature sources for each class of method cited.

- Similarly, the privacy assessment dimensions noted are described briefly, and again, some further cross-reference to supporting materials would be a welcome addition.

- Since the source data was derived from a DREAM challenge, its scope, heterogeneity, and content may not be representative of "real world" data (e.g., it is highly curated). This limitation needs to be discussed/acknowledged further.

Reviewer 1:

1. *The paper would be clarified and strengthened if they correctly broke out these three categories (i.e., resemblance, utility and privacy). It is important to do so. A single dataset can commonly be used for many purposes thus many different utilities could be assessed. A dataset can have poor resemblance but still do a good job for a specific task. If resemblance was perfect, then utility would be good for most any task. For the purpose of this review, we will consider resemblance to be different than utility and we strongly recommend that future revisions of the paper do this as well.*

Response: We thank the reviewer for suggesting a separation of the metrics characterizing data resemblance from other non-privacy metrics. We agree and have revised the manuscript accordingly. In the process, we updated Figure 1 by adding a “Resemblance” block, which is connected to the “Feature-level statistics” and “Record-level consistency” branches. This block was incorporated at the same level as the “Outcome prediction” block.

At the same time, we recognize that we used a different interpretation of the term “utility” than the reviewer. However, we relied upon the terminology as it is defined in the data privacy literature [A][B][C][D], where privacy and utility are communicated as competing goals. Additionally, in the literature utility indicates the general quantifiable benefit of data for its consumers. In this respect, the notion of resemblance is a specific realization of utility.

We revised the Data Utility Subsection to clarify this issue as follows:

“In this work, we rely upon the terminology as it has been communicated in the literature [A][B][C][D], where privacy and utility are typically posed as competing concepts and utility indicates the general quantifiable benefit of data for its consumers. In this respect, the notion of resemblance is a specific realization of utility.”

[A] Wan, Z. et al. *Expanding access to large-scale genomic data while promoting privacy: A game theoretic approach. Am. J. Hum. Genet.* 100, 316–322 (2017).

[B] Wan, Z. et al. *Using game theory to thwart multistage privacy intrusions when sharing data. Sci. Adv.* 7, eabe9986 (2021).

[C] Sankar, L., Rajagopalan, S. R. & Poor, H. V. *Utility-privacy tradeoffs in databases: An information-theoretic approach. IEEE Transactions on Information Forensics and Security* 8, 838–852 (2013).

[D] Li, T. & Li, N. *On the tradeoff between privacy and utility in Data Publishing. Proceedings of the 15th ACM SIGKDD International Conference on Knowledge Discovery and Data Mining* (2009).

2. *(a) A weakness is the final metrics are ranked-based so the results from one paper can't easily compared to the results in other papers.*

Responses: We agree with the reviewer that the final results are ranked-based and they are not directly comparable to results in other papers. However, all intermediate results from individual metrics are provided in “Supplement B: Detailed results for evaluation metrics”. These intermediate results are directly comparable to results in other papers using those metrics (e.g., meaningful identity disclosure risk). We have not found a universal metric that can perfectly aggregate all intermediate metrics. As a result, we have added the following sentences to the Discussion section: “Given this situation, there is clearly a need for universal metrics that can be compared across datasets for different models.”

*(b) To show my synthetic method is better than prior methods using this benchmark, I have to rerun from scratch all the entire methods. So, while it is certainly very useful and important to identify top models for a specific problem and their approach is clever, there is still a lot of room for **future work** in practical synthetic data benchmarking. In the end, the topped ranked methods might not be that good.*

Responses: We agree that a user has to rerun all of the methods mentioned in our paper on their own dataset to demonstrate that their new method is better than others. This is because such a comparison is dataset-specific. This is not unique to our metrics, as even F1 scores have the same problem. For example, classifier A could achieve an F1 score on dataset D1 that is higher than what classifier B achieves on dataset D2 while simultaneously achieving a lower F1 score on dataset D1 than what Classifier B achieves. This indicates that, even if all metrics are scaled (i.e., normalized and standardized) into the same range (e.g., $[0, 1]$), it would not be meaningful to directly compare them on different datasets using different methods. In summary, with a new dataset and a new method, it is always necessary to rerun all baseline methods on the new dataset in order to compare them to a new method. We have made the revision accordingly to highlight the need to propose universal metrics in the future, as stated in our answer to Question 2a.

(c) Many of the metrics are not scaled, so it is not easy to know if the score is good or bad between different datasets.

Responses: We wish to point out that some of the metrics we use are in the range of $[0, 1]$ (i.e., the three privacy risk metrics as well as the two outcome prediction utility metrics). Two metrics are in the range of $[0, 1000]$ (i.e., dimension-wise distribution) and $[0, 10^6]$ (i.e., column-wise correlation) and one metric (i.e., latent cluster analysis) is in the range of $[-\text{Inf}, \log 1]$ which corresponds to $\log([0, 1])$. These metrics are comparable across datasets given a specific method. The only metric that was not scaled to a certain range is the feature selection metric. To address this problem, we have normalized the values for this metric into the range of $[0, 1]$ (See our answer to Question 3). In addition, the two new metrics were scaled such that medical concept abundance is always in the range of $[0, 1]$ and NNAA risk is always in the range of $[-1, 1]$.

- 3. The approach is based on a suite of data utility, resemblance, and privacy metrics. They did a nice job on the task specific utility metrics which include model performance and feature selection*

metrics. The measure of shared features is very algorithm specific and dependent on the total number of features.

Response: This is an excellent point. We revised the algorithm so that it computes the proportion of shared features (i.e., the number of overlapping important features divided by the total number of important features captured from real data), instead of the counts of features. As a result, this metric is now independent of the size of the feature space. We revised the corresponding descriptions and results in the manuscript to reflect this amendment (See Table 1, Figure 3, Supplementary Figure A.2, and Supplementary Table B.7). At the same time, we wish to communicate that this revised definition did not change the ranks of the models.

4. *Other metrics might be stronger than a difference in the count – perhaps you could borrow NLP metrics for comparing two search results? Then one could tell if the results was strong or not.*

Response: We believe that our answer to Question 3 addresses the reviewer’s concern. However, if the reviewer has specific suggestions that they believe should be incorporated into the manuscript, we would welcome them.

5. *They missed the nearest neighbor adversarial accuracy metric which was designed for specifically for assessing privacy in synthetic data (see Yale 2020 papers).*

Response: We appreciate the reviewer’s suggestion and we added the nearest neighbor adversarial accuracy risk as a new privacy risk metric, called *NNAA risk* for short. We added the following paragraphs into the Methods section:

“Nearest neighbor adversarial accuracy risk. Overfitting can induce privacy risks for synthetic data. Yale *et al.* introduced a privacy loss metric for synthetic data that directly measures the extent to which a generative model overfits the real dataset based on the notion of nearest neighbor adversarial accuracy (NNAA) [E][F]. The NNAA risk we use in our evaluation is based on this metric. Specifically, let S_T , S_S , and S_E be three sets of samples with the same size from three datasets: $S_T = \{x_T^1, \dots, x_T^n\}$ from the real training dataset, $S_S = \{x_S^1, \dots, x_S^n\}$ from the synthetic dataset, and $S_E = \{x_E^1, \dots, x_E^n\}$ from the evaluation dataset. The NNAA risk is calculated as the difference between two distances:

$$AA_{ES} - AA_{TS}$$

in which

$$AA_{ES} = \frac{1}{2} \left(\frac{1}{n} \sum_{i=1}^n 1(d_{ES}(i) > d_{EE}(i)) + \frac{1}{n} \sum_{i=1}^n 1(d_{SE}(i) > d_{SS}(i)) \right)$$

$$AA_{TS} = \frac{1}{2} \left(\frac{1}{n} \sum_{i=1}^n 1(d_{TS}(i) > d_{TT}(i)) + \frac{1}{n} \sum_{i=1}^n 1(d_{ST}(i) > d_{SS}(i)) \right)$$

where the indicator function $1(\cdot)$ equals one if its argument is true and zero otherwise. $d_{TS}(i) = \min_j \|x_T^i - x_S^j\|$ is defined as the distance between $x_T^i \in S_T$, a data point in the sample from the real data, and its nearest neighbor in S_S , the sample from the synthetic data. Furthermore, $d_{ST}(i) = \min_j \|x_S^i - x_T^j\|$ is the distance between $x_S^i \in S_S$, a point in the sample from the synthetic data, and its nearest neighbor in S_T , the sample from the real data. Moreover, $d_{TT}(i) = \min_{j, j \neq i} \|x_T^i - x_T^j\|$ is the distance between x_T^i and its nearest neighbor in a sample of size $(n-1)$ instances drawn from the same distribution, and $d_{SS}(i) = \min_{j, j \neq i} \|x_S^i - x_S^j\|$ is the distance between x_S^i and its nearest neighbor in a sample of size $(n-1)$ instances drawn from the same distribution. Similarly, $d_{ES}(i) = \min_j \|x_E^i - x_S^j\|$ is the distance between $x_E^i \in S_E$, a data point in the sample from the evaluation data, and its nearest neighbor in S_S , the sample from the synthetic data. Furthermore, $d_{SE}(i) = \min_j \|x_S^i - x_E^j\|$ is the distance between x_S^i , a data point in the sample from the synthetic data, and its nearest neighbor in S_E , the sample from the evaluation data. Moreover, $d_{EE}(i) = \min_{j, j \neq i} \|x_E^i - x_E^j\|$ is the distance between x_E^i and its nearest neighbor in a sample of size $(n-1)$ drawn from the same distribution.

The original NNAA risk as defined by Yale *et al.* [E] requires all samples of datasets to have the same size; however, in our experiments, the sizes of the training and synthetic dataset are both larger than the size of the evaluation dataset. Thus, we randomly sample the training dataset and the synthetic dataset to be in the same size of the evaluation dataset multiple times and use the average result as the NNAA risk. In addition, we normalize all continuous attributes into a $[0,1]$ range before computing distances.”

In addition, we modified Figures 4 and 5, Table 4, Supplementary Figure A.3, and Supplementary Tables A.1-4, B.11, and D.1, as well as the corresponding descriptions.

[E] Yale, A. *et al.* *Generation and evaluation of privacy preserving synthetic health data. Neurocomputing* 416, 244–255 (2020).

[F] Yale, A. *et al.* *Synthesizing quality open data assets from Private Health Research Studies. Business Information Systems Workshops* 324–335 (2020). doi:10.1007/978-3-030-61146-0_26

6. *It's great that they included a clinical knowledge violation metrics. When available this is a great thing to do. But what about all the other inaccurate relationships that the synthetic data introduce? You need more general resemblance metrics that will capture this. One possibility is some metric based on sums of phecodes per patient. MEDGAN was found to generate unrealistic patients with many diagnosis codes in Yale 2020.*

Response: The reviewer’s suggestion certainly enhances the benchmarking framework with respect to the dimension of record-level data quality evaluation. As such, we integrated a new metric called *medical concept abundance* to compute the normalized Manhattan distance between the histograms of the number of assigned distinct medical concepts for real patients and synthetic patients. Specifically, we fix the space to the total number of distinct medical concepts that are considered in synthesis and then divide the range evenly into M bins. We then compute $\sum_{i=1}^M |h_r(i) - h_s(i)|/2N$, where $h_r(i)$ and $h_s(i)$ represent the number of patients in the i^{th} bin that are real and synthetic, respectively, and N represents the total number of real patients. The metric

is thus in the [0,1] range and characterizes the degree to which a synthesis model captures the quantity of record-level information in the real data. We now describe this metric in the Method Section (see below) and updated Figures 1, 3 and 5, Tables 1 and 4, along with relevant content in the supplemental materials, accordingly. The key takeaway message is that EMR-WGAN outperforms the other models because it achieves the lowest medical concept abundance distances. This is consistent with the findings from the other utility metrics.

“Medical concept abundance. Close resemblance in feature-level metrics does not necessarily imply high similarity in the record-level distributions between the real and synthetic datasets. Thus, inspired by the work of Yale and colleagues [E], we introduce a metric that characterizes the degree to which a synthesis model captures the quantity of record-level information in the real data. Specifically, we compute the normalized Manhattan distance between the histogram of the number of assigned distinct medical concepts for real and synthetic records. To do so, we fix the space to the total number of distinct medical concepts that are considered in synthesis and then divide the range into M evenly-sized bins according to the desired assessment granularity. We then computed the medical concept abundance distance as $\sum_{i=1}^M |h_r(i) - h_s(i)|/2N$, where $h_r(i)$ and $h_s(i)$ represent the number of records in the i^{th} bin that are real and synthetic, respectively, and N denotes the total number of real records. This metric is thus in the [0,1] range, where a lower the value indicates a higher real-synthetic data similarity in terms of record-level information distributions. In this study, we set M equal to 20.”

[E] Yale, A. et al. *Generation and evaluation of privacy preserving synthetic health data. Neurocomputing 416, 244–255 (2020).*

7. *The other choices of resemblance metrics seem necessary but not sufficient. Mode collapse is a huge problem with GANs so subgroups of the dataset may be lost. GANs are tricky to train. In Figure 2, DPGAN is clearly failing catastrophically. The univariate marginal metric: dimension wise distribution metric, is okay – but taking average or Wasserstein distances on very high dimensional can mask the fact that rare data classes distribution are altered from real to synthetic data. The remaining multivariate metrics seem lacking. Column wise correlation only captures simple linear correlations. The Latent cluster analysis was a good idea but, in the end, only three clusters were used. This could easily miss mode collapse. Some other metrics that can make sure that all the training data has a reasonable synthetic counterpart (but not too close) and that vice versa are needed sorely needed. Recent studies have shown that synthetic data can introduce significant biases when resemblance and fairness is evaluated with respect to critical covariates (Bhanot 2021).*

Response: The reviewer is correct that GAN models are susceptible to mode collapse, which, in turn, can induce poor synthetic data quality. In this respect, we acknowledge that the three clusters applied in latent cluster analysis in this study might not be sufficient to reveal the presence of mode collapse. However, we do believe that latent cluster analysis can address this problem. This is because the inspection granularity can be controlled by specifying the number of clusters in the

clustering algorithm. Thus, in the revised manuscript, we rewrote the following sentence for the “Latent cluster analysis” paragraph of the Methods Section:

“We then applied k -means to define the clusters, where k was determined according to the elbow method [F] (which was found to be three for both datasets in this study). It should be recognized that the elbow method is a heuristic and the number of clusters could alternatively be specified according to the desired granularity of quality inspection.”

We also thank the reviewer for raising the fairness issue, which is certainly an important topic for synthetic data evaluation. However, we believe this is a complex problem that is still very much evolving and, thus, leave it as an opportunity for future work. In this respect, we added the following sentence (as one of the limitations) at the end of the Discussion Section:

“Sixth, there is mounting evidence that suggests generative models may induce bias and fairness issues, such that subgroups of the population are not evenly well generated [G]. As a consequence, the resulting synthetic data, in certain circumstances, could accentuate disparities in health. We believe that the benchmarking framework introduced in this paper would benefit from extensions to cover bias and fairness dimensions. However, measuring bias and fairness in synthetic data is still very much an open problem and likely requires further investigation before a canonical set of metrics are ready for integration.”

[F] Yuan, C. & Yang, H. *Research on k -value selection method of k -means clustering algorithm*. *J* **2**, 226–235 (2019).

[G] Bhanot, K., Qi, M., Erickson, J. S., Guyon, I. & Bennett, K. P. *The problem of fairness in Synthetic Healthcare Data*. *Entropy* **23**, 1165 (2021).

8. *Are the observed differences significant? Could you incorporate statistical significance?*

Response: The statistical significance of the observed differences for each metric cannot be tested in many cases due to the fact that there were only three runs of models. This would be possible if additional runs were performed, but each run required substantial computational resources such that there is no simple cost-effective way to feasibly do so at the present moment in time. Research is needed to figure out how to speed up the process or make it cost-effective. Still, our rank-based method does not rely on statistical significance to derive model comparison. We added the following sentence to the Limitations and future work section:

“Eighth, the statistical significance of the observed differences for each metric cannot be tested in many cases due to the fact that there were only three runs of models. This would be possible if additional runs were performed, but each run required substantial computational resources such that there is no simple cost-effective way to feasibly do so at the present moment in time. However, our rank-based method does not rely on the statistical significance to derive model comparison.”

9. *In Figure 2, DPGAN and WGAN on UW, it looks like DPGAN and WGAN are taking very rare features and making them very common. This shows a catastrophic failure of these two methods and is a bit unusual for a GAN result. Could you improve tuning? Also, it is possible that y is real and x is synthetic in Figure 2? Then this would be consistent with a mode collapse which is quite common. Probably worth a comment on why the methods are doing so poorly. Also, you used a huge value of epsilon so no theoretical proofs of privacy here.*

Response: This is an excellent question. First, the y -axis corresponds to synthetic data. The synthetic datasets generated by DPGAN and WGAN (on the UW dataset) were indeed skewed by a larger set of binary features gaining higher prevalence than in the real data. This was actually something that we recognized and investigated before submitting the initial manuscript. We trained the two models 20 times (each took around two GPU days) with various tuning procedures and monitored the log of GAN losses. However, this phenomenon was consistently observed even though the losses were as expected. We believe the distributional complexity of the UW dataset, in combination with the mechanisms of WGAN, mainly contributed to this phenomenon. Additionally, adding noise to the training process (i.e., DPGAN) only accentuated the poor performance. As a result, it is evident that the GAN training process could lead to mode collapse in the form of an underrepresentation of features, but also overrepresent the features. We have revised the manuscript in the Discussion Section accordingly:

“Notably, with respect to APD (Fig. 2), while WGAN and DPGAN perform well for the VUMC dataset, they do not for the UW dataset. This may stem from the differences in the complexity of the joint distribution between the two real datasets and the WGAN mechanism applied to this dataset (note that DPGAN was implemented based on WGAN).”

We agree with the reviewer that a smaller epsilon will provide stronger privacy protection. However, we chose to report the results with the current selection of epsilon value (which is relatively large) for several reasons. First, the epsilon value we utilized leads to very poor utility for DPGAN, in terms of resemblance and outcome prediction (Fig. 5A-B). A typically small epsilon will make DPGAN perform much worse, such that the generated data has no value. Importantly, we believe the premise of differential privacy is to support a trade-off between utility and privacy, rather than entirely sacrificing utility. Second, the privacy risks for the GAN models were evaluated through measures that are not required to comply with DP guarantees.

10. *The section on synthetic paradigms is not very clear. Synthetic paradigms were not defined earlier in the paper. It only works on one of the datasets. The results are primarily in the supplement. The topic seems kind of gratuitous and not the main subject of the paper. The contribution is not clear. The paper is about benchmarks not better methods. Recommend keeping for a later paper. Hopefully, your new improved benchmarks will stimulate lots of better reproducible synthetic data generation methods in the future.*

Response: We agree and thank the reviewer for pointing out this issue and the encouraging feedback.

A synthesis paradigm is defined as the way a generative model utilizes core features in a real dataset, which could be the key outcome or demographic feature, to generate synthetic data. For example, one can use a *separated paradigm* (i.e., train two GAN models to generate the positive and negative cohorts separately), a *combined paradigm* (i.e., train only one GAN model to generate both cohorts by treating the outcome feature as a regular column the same as others), or other conditional paradigms (where given features are taken as conditions of GAN training) to train models.

We would like to keep this as an important component of the framework for several reasons. First, the synthesis paradigm is a concept that is orthogonal to the selection of GAN models. In addition to selecting a generation model, the people who train a synthesis model need to select a synthesis paradigm. Second, given a real dataset, the privacy and utility evaluation results of data synthesis can vary due to differences in their synthesis paradigm (as shown in Supplementary Table A.4). Thus, determining the optimal pair of synthesis paradigm and GAN model is critically important. Our benchmarking framework accommodates this complexity.

We focused on two candidate synthesis paradigms with the UW Mortality dataset for illustration purposes. We applied only one synthesis paradigm (i.e., the combined paradigm) on the VUMC COVID Admission dataset because the positive class in the cohort is too small (~500 patients) to support separated GAN training (as communicated in the Methods section). The main reason we moved some of the results to the Supplementary Tables/Figures is an artifact of the journal's limit on the number of display items.

To provide greater clarity, we rewrote the first paragraph of the Synthesis Paradigm Subsection as follows:

“A data synthesis paradigm corresponds to the manner by which a generative model utilizes features in a real dataset to generate synthetic data. These features could be a key outcome (e.g., a readmission event) or demographics (e.g., age of the patient). The selection of a synthesis paradigm has an impact on the utility and privacy of synthetic data. It should be noted that selecting synthesis paradigms and candidate generative models are two facets of the synthesis process embedded into the Synthetic EHR data generation phase (Fig. 1). The benchmarking framework was thus designed to accommodate the need to incorporate different data synthesis paradigms according to the key features (e.g., the 21-day hospital admission post COVID-19 positive testing and six-month mortality in general) as part of the benchmarking. In this study, we specifically evaluated two synthesis paradigms: 1) the *combined synthesis paradigm* (Fig. 6C) and 2) the *separated synthesis paradigm* (Fig. 6D). Further details are provided in Methods.”

11. *For membership inference risk, the description could be clearer. “we assume that the adversary is in possession of all attributes of a set of targeted records and we know whether each targeted record is in the real dataset” is not clear. Who is we? Does this include the adversary? Does the attacker have the full set of synthetic data as well. Does the attacker leverage the synthetic data?*

Shouldn't you evaluate the synthetic data as well? The work on Nearest Adversarial Accuracy could be useful here.

Response: “We” means risk evaluators (i.e., users of the benchmarking framework) including the authors. Without making this assumption, we cannot compute the F1-score, which is the performance of the adversary’s attack. This assumption is necessary for a risk evaluation with an adversary model.

In this respect, it should be made evident that “we” does not include the attacker. Rather, we assume the attacker has access to and leverages the entirety of the synthetic data. Thus, our evaluation focuses on the synthetic data.

We revised the manuscript to read: “we assume that the adversary is in possession of the synthetic data and all attributes of a set of targeted records. We further assume that risk evaluators (i.e., users of the benchmarking framework) know whether each targeted record is in the real dataset”.

We have added the privacy metric derived from the concept *Nearest Neighbor Adversarial Accuracy* in the revised manuscript. Please see our answer to Question 5 above.

Reviewer 2:

- 1. The summary of generative methods is rather cursory. While this is not the core premise of the paper, it is important, and many readers may not be familiar with such approaches. One possible solution to this would be the addition of a table or other construct that could summarize current (recommended) literature sources for each class of method cited.*

Response: We agree and added the following table to the Results section. Additionally, we provide detailed descriptions of these models in the Methods section.

Table 2. A summary of GAN models considered in benchmarking. All models share the generator-discriminator architecture but differ in their own design to enhance utility or privacy of EHR data synthesis.

Model	Distance measure (loss function)	Auto-encoder for discrete data generation	Normalization	Additional privacy design
medGAN ²⁵	Jensen-Shannon divergence	Yes	BatchNorm for generator	No
medBGAN ³⁴	f-divergence	Yes	BatchNorm for generator	No
EMR-WGAN ²⁰	Wasserstein divergence	No	BatchNorm for generator; LayerNorm for discriminator	No

WGAN ³⁴	Wasserstein divergence	Yes	BatchNorm for generator	No
DPGAN ³⁵	Wasserstein divergence	Yes	BatchNorm for generator	Yes (differential private stochastic gradient descent)

2. *The privacy assessment dimensions noted are described briefly, and again, some further cross-reference to supporting materials would be a welcome addition.*

Response: We followed the reviewer’s suggestions by 1) revising the introduction of utility and privacy metrics in Table 1 in a semi-structured manner such that the core factors of each metric can be understood quickly, 2) adding relevant references, and 3) highlighting in the first paragraph of the Benchmarking framework subsection that “Detailed metric descriptions can be found in Methods.”

3. *Since the source data was derived from a DREAM challenge, its scope, heterogeneity, and content may not be representative of "real world" data (e.g., it is highly curated). This limitation needs to be discussed/acknowledged further.*

Response: We appreciate the reviewer for raising this important issue. As suggested, we added the following paragraph as a limitation of this study in the Discussion Section of the revised manuscript.

“Fifth, the two datasets utilized in our evaluation, though derived from real EHR systems, were curated through a series of preprocessing steps. As a result, they may not represent the full scope of EHR data complexity. For instance, they may not address heterogeneity in data modality, organizing structures, missingness patterns, and the size of feature space, among other aspects. As such, further investigation is needed to examine, and potentially extend, the applicability of the benchmarking framework for real-world datasets with various properties.”

Reviewers' Comments:

Reviewer #1:

Remarks to the Author:

The authors did a good job of responding to the reviewers comments and the paper is suitable for publication.

Reviewer #2:

Remarks to the Author:

The authors have responded to the prior critique in a comprehensive manner and the ensuing manuscript should make an important contribution to the state-of-knowledge concerning the evaluation of synthetic data generation methods.

Reviewer 1:

- 1. The authors did a good job of responding to the reviewers comments and the paper is suitable for publication.*

Response: We thank the reviewer for the positive feedback and all previous suggestions that improved this manuscript.

Reviewer 2:

- 1. The authors have responded to the prior critique in a comprehensive manner and the ensuing manuscript should make an important contribution to the state-of-knowledge concerning the evaluation of synthetic data generation methods.*

Response: We thank the reviewer for the encouraging comments and helpful suggestions during the whole review process.